# MODELING VISUAL CORTEX BY MAXIMIZING LAYER-WISE MULTISCALE MANIFOLD CAPACITY

## ABSTRACT

Task-optimized deep neural networks have risen to prominence as the most predictive phenomenological models of responses in primate visual cortex, but leave much to be desired from the perspective of biological plausibility. One such limitation is the reliance on precise credit assignment through global backpropagation of error signals. Recent work has shown that this weakness can be circumvented by requiring each subsequent stage to solve a distinct and increasingly complex task, allowing for layerwise local learning signals. We propose a novel strategy for crafting such intermediate losses that uses an efficient coding framework formulated in terms of manifold capacity, which can be computed using a sequence of canonical cortical computations. In particular, we leverage the relationship between the multiscale nature of visual signals and the dilation of receptive field sizes in cascaded visual representations to modulate complexity, allowing for the reapplication of these common loss computations at each stage of the hierarchy. We evaluate our approach on its ability to predict neural datasets spanning three areas of the ventral stream hierarchy in macaques, as well as human psychophysical data on an object classification task. We find that our unsupervised layerwise model matches or exceeds the performance of competitive architecture-matched baselines on all evaluations considered.

## 1 INTRODUCTION

The ventral stream of the primate visual cortex has served as a reference for object recognition for many decades. Traditional models, based on linear filtering, rectification, and gain control have accounted for response properties of neurons in the early stages of this stream (e.g., areas V1 and V2, Heeger (1992); Felsen et al. (2005); Vintch et al. (2015); Liu et al. (2016); Willmore et al. (2010)), but have proven difficult to generalize to later stages (e.g., areas V4 and IT). Deep neural networks (DNNs) currently offer the strongest predictive models of neural responses in these later stages Yamins et al. (2014); Zhuang et al. (2021); Willeke et al. (2023), but these results come with a number of limitations. For one, their predictions are obtained by regressing a large number of model neurons (typically thousands) onto each biological neuron, risking overfitting and allowing for a generic nonlinear approximation with minimal interpretability or insight into biological properties. Additionally, most DNN training aims to optimize an end-to-end objective function, which requires precise credit assignment via global backpropagation of error signals. This computation is generally thought to be implausible for biological implementation, and it may also provide insufficient constraint on the internal representation of the network, especially at the earlier stages of processing. In light of this, it is perhaps unsurprising that many of the best recent DNN models of early cortical representations are those that incorporate additional constraints. In particular, both adversarial training (which enforces robustness to small image perturbations - Madry et al. (2018)), and layerwise learning (which directly applies objective functions at multiple stages of a network's hierarchy - Parthasarathy et al. (2024)), have offered improvements.

Here, we develop a layerwise self-supervised learning strategy that addresses the limitations of end-to-end trained DNNs, while avoiding the high computational cost associated with adversarial learning. The objective for each layer is a particular form of coding efficiency, which seeks a compromise between representational quality and resource limitations. For the former, we use *manifold capacity*, which quantifies the number of neural manifolds that can be linearly separated in a population of neurons (Chung et al., 2018). Recent work has successfully adapted this framework into a self-

supervised learning objective(Yerxa et al., 2023). For the latter, we constrain the computational capacity of each stage of processing through the network architecture. Early layers have fewer neurons with smaller receptive fields, and are also less expressive in terms of the complexity of their response functions (because they are computed with a smaller set of rectification operations). We also seek to match the complexity of the learning task (or measure of representational quality) to the computational capacity at each stage where the loss is applied (Parthasarathy et al., 2024), by applying each layer's objective to a localized region whose size is proportional to the architecturally-constrained receptive field size. This results in a canonical implementation of the loss function at each stage of the network hierarchy, which can be computed using operations consistent with response properties observed in cortical neurons.

We implement a three-stage network based on the AlexNet architecture, and train each stage independently using synthetic videos derived by sampling still images from the ImageNet dataset and simulating their evolution over time with a randomly chosen mixture of translation, dilation, and intensity shifting operations. The self-supervised MMCR objective is applied to the temporal responses of each layer, over intervals whose duration also scales with the corresponding receptive field size. We show that the resulting network outperforms an architecturally-matched object-recognition network, as well as its adversarial robust extension, in explaining responses of primate neurons recorded in areas V1, V2, and V4. Moreover, this improvement in performance holds when restricting the regression fit to a smaller set of model neurons. Finally, we compare our trained network to human performance on classification tasks, and find that it supports downstream object classification behavior that is more robust to distribution shifts and better aligned with human behavior than networks trained supervised or adversarial learning.

## 1.1 RELATED WORK

In the context of modern machine learning, layerwise or "greedy" learning algorithms were initially developed as pre-training methods whose goal was to provide a good initialization for eventual end-to-end optimization (Bengio et al., 2006; Hinton et al., 2006). More recent work has demonstrated that the composition of gradient-isolated modules can achieve respectable levels of performance on downstream tasks when employing either supervised or self-supervised learning objectives (Löwe et al., 2019; Belilovsky et al., 2019; Siddiqui et al., 2024). In computational neuroscience, there has been a longstanding interest in developing and analyzing learning algorithms that update synaptic strengths based on local signals (e.g., Hebbian learning rules). Two recent examples that apply such rules to hierarchical representation learning are Contrastive Local and Predictive Plasticity (CLAPP - Illing et al. (2021)) and Layerwise Predictive Learning (LPL - Halvagal & Zenke (2023)). By and large these algorithmic advances have fallen short of quantitatively improving our ability to model the responses of real neurons.

Parthasarathy et al. (2024) argue that one key limitation of prior approaches is the failure to align task complexity with the representational capacity of each layer, which naturally increases with depth. By explicitly matching complexity to capacity, they achieve state-of-the-art neural predictivity in area V2. However, their implementation relies on providing each layer with a distinct input stream, modulating the augmentation strength used to train each stage via self-supervision. While intuitive, this strategy strains the analogy between model-training and any biological optimization algorithm which must of course use a single stream of visual inputs to learn all stages.

## 2 METHODS

Deep convolutional networks transform an input image $x \in \mathbb{R}^{c \times h \times w}$ into a sequence of internal activation maps that preserve the 2-D topology of the input $\{a^{(1)} \in \mathbb{R}^{c_1 \times h_1 \times w_1}, ..., a^{(n)} \in \mathbb{R}^{c_n \times h_n \times w_n}\}$ through a cascade of linear-nonlinear operations (i.e. $a^{(1)} = f_{\theta_1}(x), a^{(2)} = f_{\theta_2}(a^{(1)})$, and so on). At each stage the receptive field of a single unit (the region of the input $x$ to which $a^{(i)}$ is sensitive) grows due to the repeated application of convolutional filters and downsampling operations in each stage. The response of neurons in deeper stages are thus sensitive to larger and more complex visual features, a property also found in neurons of successively deeper stages of the primate ventral stream. This is well matched to the *expressivity* growth of subsequent stages: $f_{\theta_2} \circ f_{\theta_1}$ can express more complicated functions than $f_{\theta_1}$ alone by virtue of the additional trainable parameters in $f_{\theta_2}$ and a second point-wise nonlinearity.

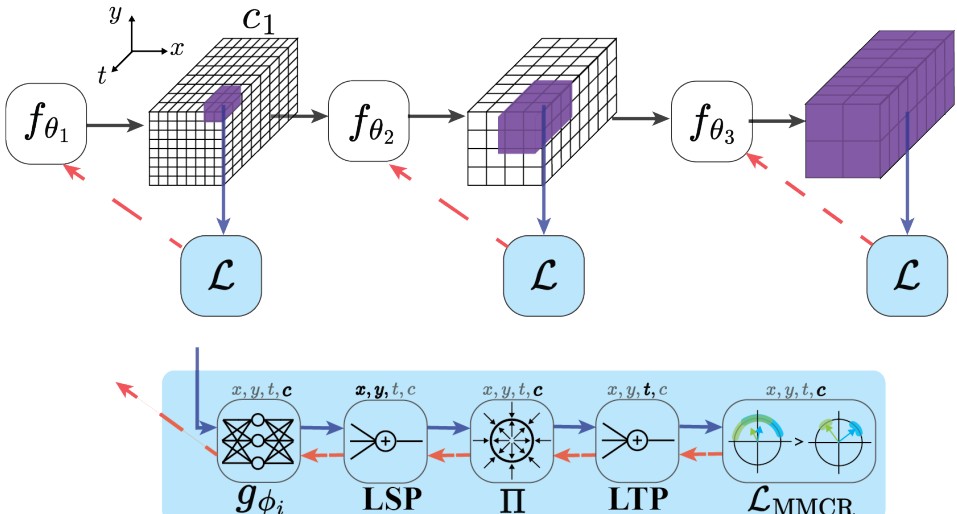

Figure 1: **The ST-MMCR training scheme.** In response to a video input, each stage of the network produces spatio-temporal feature maps ("channels", $c_j$, one shown for each stage, as a 3D block), with successively reduced spatial resolution, but fixed temporal resolution. The objective that is used to optimize parameters $\theta_i$ for the $i$th stage operates on a spatio-temporal region (indicated in purple) whose extent in both space and time successively increases by factors of two at each stage. This objective is of identical form at each stage, and consists of a sequence of computations (exploded diagram at bottom), each operating over one or two indices (indicated in bold above each computation): 1) a multi-layer perceptron $g_{\phi_i}$ with a single hidden layer is applied across channels, at each spatio-temporal location; 2) these responses are spatially pooled (averaged) over a local central region (purple block); 3) these are normalized (i.e., divided by their Euclidean length) over channels; 4) these are temporally pooled to yield manifold centroids; 5) the MMCR loss (negative nuclear norm of the matrix of centroids) is computed.

Dilating receptive field sizes and computational expressivity can be leveraged to learn invariances to more complicated features and transformations at subsequent stages through the application of distinct loss functions at each layer. This is the key idea behind the Layerwise Complexity-matched Learning (LCL) scheme of Parthasarathy et al. (2024), where image patches of different sizes and subjected to a variety of transformations of different strength are used to train two stages of a network independently (with layerwise losses and without backpropogation between stages). However in biological vision, the inputs to each stage of the hierarchy arise from a common visual stimulus, and that input arrives in a continuous temporal stream, rather than as samples randomly transformed relative to a base image. In this work we address this mismatch by introducing an objective based on local spatial pooling, and integration over time. Both the spatial pooling region and the temporal duration increase with each successive layer, in proportion to the receptive field sizes.

## 2.1 AN ARTIFICIAL VIDEO TRANSFORMATION

Real-world video data would provide the most natural and ecologically valid approach for training our models, but they present challenges for fair comparison and experimental control. A large body of models – particularly those pretrained on ImageNet – derive their inductive biases from the statistics of static natural images, and shifting to real video datasets introduces substantial variation in the distribution of visual content (which can confound downstream model comparisons (Conwell et al., 2022)). Instead, we train our models on simulated video sequences with smoothly evolving transformations. Starting with a single image randomly selected from the ImageNet dataset, we take two random (resized) crops and linearly interpolate their crop parameters to produce a sequence of frames with gradually shifting viewpoints. Following Parthasarathy et al. (2024) we additionally apply mild random photometric transformations (contrast and brightness modulation, stochastic conversion between color and grayscale, and the addition of gaussian noise), to each frame, and omit the more aggressive augmentations (e.g., hue shifts, solarization etc.) used in many SSL training schemes (Grill et al., 2020). For full details see A.2. These stimuli preserve alignment with

ImageNet-pretrained models, allowing for controlled comparisons across training paradigms. Furthermore, this strategy enables fine-grained parametric control over the complexity of spatiotemporal transformations, providing a powerful tool for probing the relationship between transformation structure and representational learning.

## 2.2 ARCHITECTURE

Following Parthasarathy et al. (2024), we adopt AlexNet with batch normalization as our backbone architecture (Krizhevsky et al., 2012; Ioffe & Szegedy, 2015). While more recent architectures allow for large gains in performance on computer vision tasks, they provide minimal improvement in terms of neural predictivity, and the simplicity of the AlexNet architecture (a cascade of linear-nonlinear downsampling operations) strengthens the analogy between the model's computations and the initial feedforward cascade of transformations observed in the ventral stream (El-Shamayleh et al., 2013; Ziemba et al., 2016).

Our network is constructed as a cascade of three stages $\{f_{\theta_1}, f_{\theta_2}, f_{\theta_3}\}$, corresponding to primate visual areas V1, V2, and V4, respectively (see Fig. 1). The first two stages consist of a convolution, a halfwave rectifying (ReLU) nonlinearity, batch normalization, and a Max Pooling operation (which includes spatial downsampling by a factor of two). Thus the computational capacity of $f_{\theta_2} \circ f_{\theta_1}$ is approximately double that of $f_{\theta_1}$. To double the relative capacity again, $f_{\theta_3}$ includes 2 stages of linear-nonlinear operations and concludes with a final downsampling operation. Note that this slightly deviates from the original AlexNet architecture which has 3 linear-nonlinear blocks before the third downsampling layer. The MMCR loss function is computed through additional nonlinear "projection heads", as is standard in SSL. Following Parthasarathy et al. (2024), each of these projection heads (denoted $g_{\phi_1}$, $g_{\phi_2}$ and $g_{\phi_3}$) are implemented with 1-hidden layer MLPs.

## 2.3 OBJECTIVE

**The MMCR objective function.** Contrastive self-supervised learning methods aim to learn a representation, $f_\theta$, that is invariant to some set of random input transformations, $\tau_\rho$, and simultaneously discriminative across distinct inputs (Chen et al., 2020b; Zbontar et al., 2021; Bardes et al., 2022; Yerxa et al., 2023). Maximum manifold capacity representations (MMCR) cast SSL as capacity optimization and aim to maximize the number of linearly separable "transformation manifolds," that can be stored in the representation space produced by $f_\theta$ (Yerxa et al., 2023). For each input image in a dataset (notated as a vector $\mathbf{x}_b \in \mathbb{R}^{\mathbb{D}}$) we generate samples from the corresponding transformation manifold by applying the random transformation $k$ times, yielding manifold sample matrix $\tilde{\boldsymbol{X}}_b \in \mathbb{R}^{D \times k}$. Each transformed image is mapped to a $d$-dimensional response space by $f_\theta$ and projected onto the unit sphere yielding manifold response matrix $\boldsymbol{Z}_b \in \mathbb{R}^{d \times k}$. The centroid $\boldsymbol{c}_b$ of the transformation manifold is then approximated by taking the sample mean (averaging across the $k$ columns). For a set of images $\{\mathbf{x}_1, ..., \mathbf{x}_B\}$ we compute normalized response matrices $\{\boldsymbol{Z}_1, ..., \boldsymbol{Z}_B\}$ and assemble their corresponding centroids into matrix $\boldsymbol{C} \in \mathbb{R}^{d \times B}$.

Given the responses and their centroids, the MMCR loss function can be written simply:

$$\mathcal{L}_{\text{MMCR}} = -||\boldsymbol{C}||_*, \tag{1}$$

where $|| \cdot ||_*$ indicates the nuclear norm. Minimizing this objective simultaneously encourages transformation invariance and image-to-image discriminability, the two key conceptual ingredients in SSL, through a single term Yerxa et al. (2023). This is because individual centroids, as averages of unit vectors, have norms that are bound above by 1, and are larger when the samples from the transformation manifold are more aligned (i.e., the objective encourages invariance). Additionally, maximizing the nuclear norm encourages distinct centroids to be as near orthogonal to each other as possible (encouraging discriminability). While many SSL objective functions have been proposed, most require pairwise comparisons between the transformed inputs, and thus have complexity that is quadratic in the number of number of samples from the transformation manifold, $k$. MMCR has the unique distinction of having constant complexity: the size of $\mathbf{C}$ is independent of $k$. While this is of minimal importance in standard SSL settings where $k = 2$, our learning scheme relies on the use of large $k$, for which MMCR is uniquely well suited. It is worth noting that MMCR is not a direct optimization of capacity in general (Chung et al., 2018; Chou et al., 2024), but introduces a set of

simplifying assumptions, such as elliptically symmetric manifold geometries, in order to arrive at a tractable objective function that bears some similarity to biological computations.

**Local spatiotemporal MMCR.** Rather than random samples from a transformation manifold, we train our network using transformations that smoothly vary across $T$ frames (see Section 2.1). This choice allows us to derive a more ecological learning signal arising from *temporal and spatial proximity*. For input videos $\mathbf{x}_b \in \mathbb{R}^{T \times c \times h \times w}$ we compute corresponding feature maps produced by our 3 network stages $\mathbf{a}^{(i)} \in \mathbb{R}^{T \times c_i \times h_i \times w_i}$. To modulate the complexity of the learning task assigned to each network stage, we adjust the spatial and temporal extent of transformation manifolds on which the MMCR objective operates (see Fig. 1). We define $\mathrm{LSP}(s)$ as a local spatial pooling operation that averages over the central $s$ pixels of a feature map. Similarly, let $\mathrm{LTP}(t)$ be a local temporal pooling operation that averages over the central $t$ frames. For each input video $\mathbf{x}_b$, each network stage produces a single centroid:

$$\mathbf{c}^{(1)} = \mathrm{LTP}(\frac{t_f}{4}) \circ \Pi \circ \mathrm{LSP}(s_f) \circ g_{\phi_1} \circ f_{\theta_1}(\mathbf{x}_b) \tag{2}$$

$$\mathbf{c}^{(2)} = \mathrm{LTP}(\frac{t_f}{2}) \circ \Pi \circ \mathrm{LSP}(s_f) \circ g_{\phi_2} \circ f_{\theta_2} \circ f_{\theta_1}(\mathbf{x}_b) \tag{3}$$

$$\mathbf{c}^{(3)} = \mathrm{LTP}(t_f) \circ \Pi \circ \mathrm{LSP}(s_f) \circ g_{\phi_3} \circ f_{\theta_3} \circ f_{\theta_2} \circ f_{\theta_1}(\mathbf{x}_b) \tag{4}$$

where $\circ$ denotes function composition, and $\Pi$ normalizes vectors over channels (projecting them onto the unit hypersphere). Note that the temporal pooling in each stage operates over successively longer durations. The spatial pooling operates over different *effective* window sizes, since the channels of each successive stage are sampled at half the resolution of the previous stage. Finally, the MMCR losses for each stage are computed from the nuclear norm of the centroid matrix obtained from a batch of input videos, and summed: $\mathcal{L} = \mathcal{L}_{\mathrm{MMCR}}(\mathbf{C}^{(1)}) + \mathcal{L}_{\mathrm{MMCR}}(\mathbf{C}^{(2)}) + \mathcal{L}_{\mathrm{MMCR}}(\mathbf{C}^{(3)})$. The parameters of each layer $\{\theta_i, \phi_i\}$ are adjusted using only gradients of the corresponding stage, $\nabla_{\theta_i, \phi_i} \mathcal{L}_{\mathrm{MMCR}}(\mathbf{C}^{(\mathbf{i})})$, with no backpropagation of error signals across stages.

We optimized the aforementioned architecture using this objective function, with videos generated from base images drawn from the ImageNet-1k dataset. For the sake of computational efficiency we used the minimum allowable 8 frames and choose the pooling region sizes to be global at the output of the final stage ($T = t_f = 8$ and $s_f = 7$). For more optimization details see Appendix A.3.

## 2.4 Evaluating Neural Predictivity

**Evaluation methodology.** We evaluate models on their ability to predict trial averaged firing rates via linear regression from their internal representations. Following Schrimpf et al. (2018), we use partial least squares (PLS) regression with 25 components to map from model feature maps to predicted mean firing rates. We compute the Pearson correlation coefficient between the predicted and observed responses for each neuron, and use the median over neurons as the "score" for a given evaluation. Finally, we use k-fold cross validation and report the average score over test splits (for details see A.1).

**Physiology datasets.** For areas V1 and V2 we use neural recordings from Freeman et al. (2013); Ziemba et al. (2016). The dataset contains single unit responses for 102 V1 neurons and 103 V2 neurons to texture images and their spectrally matched noise counterparts (with a total of 450 images). For area V4 we use recordings from Lieber et al. (2024). The dataset contains responses of 211 single units in V4 that were evoked by natural images subjected to a parametrically controlled degree of "texturization," (with a total of 480 images). For further details on each dataset, see A.1.

## 3 Neural Alignment

We begin by replicating and extending the core result of Parthasarathy et al. (2024), to verify the complexity-matching justification for our architecture and objective. We trained networks using global average pooling at each stage, while varying the spatial region (stimulus size) and the duration of the training videos (which determines the amplitude of the applied spatial transformations - see A.2 for details on the augmentation procedure). We find that the successive stages of the model are

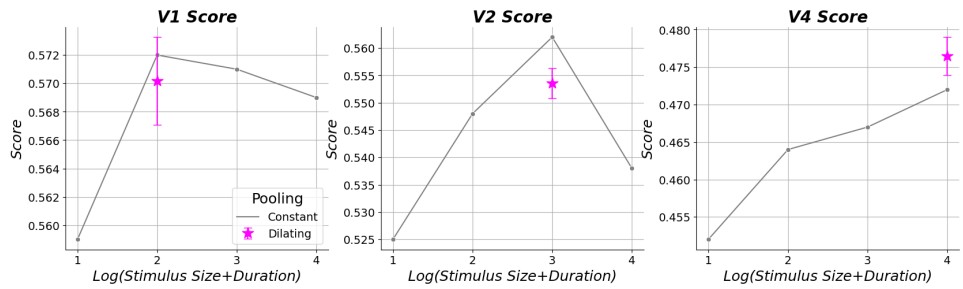

Figure 2: **Neural predictivity as a function of stimulus complexity.** Following Parthasarathy et al. (2024) we examine perfomance of a network trained using "constant" (global) spatial and temporal pooling, and modulate task difficulty by varying the size of input images and the strength of the spatial transformations they are subjected to (which is analgous to video duration). Our layerwise model, with pooling size/duration proportional to receptive field sizes in each stage, reaches near-peak performance in explaining data in all three cortical areas (red star), when trained over a single set of full-size video inputs.

best at explaining data from their corresponding cortical area when the stimulus size and duration are proportional to their receptive field size (Fig. 2). In comparison, when a single network is trained on a single set of full-size videos using stage-adapted ("dilated") pooling, which modulates the task complexity at each layer directly through the architecture, it achieves (approximately) peak performance in explaining data in all three cortical areas.

Next, we compare the neural predictivity of our model with that of a set of architecture and data-diet matched models, as well as handcrafted baseline models of V1 and V2 (see Fig. 3). Specifically, we consider the following baselines:

- **Supervised:** An AlexNet model trained with standard supervised cross entropy loss on ImageNet-1k. We use the standard weights from the pytorch library.

- **Robust:** An AlexNet model trained with L2-adversarial learning (Madry et al., 2018) that has been shown to provide a strong model of early cortical responses (ranking $3^{rd}$ of 450 models in the Brain-Score leaderboard for this V2 dataset) (Parthasarathy et al., 2024; Schrimpf et al., 2018). We use model weights obtained from (Chen et al., 2020a).

- **Random:** An AlexNet with randomly initialized weights.

- **SteerPyrV1:** A 5-scale 4-orientation steerable pyramid decomposition (Simoncelli & Freeman, 1995), augmented with simple and complex cell nonlinearities which rectify the (components of) the complex-valued coefficients and compute the complex modulus, respectively.

- **SteerPyrV2:** Complex cell outputs of the SteerPyrV1 module are subjected to a second stage of steerable pyramid filtering and complex-cell nonlinearities. This is an instantiation of a second-order scattering transform (Bruna & Mallat, 2013).

First we note that these results reproduce several key results from previous literature: (1) handcrafted filters that are tuned for orientation and spatial frequency substantially outperform learned models on the V1 dataset; (2) for V1, supervised training provides only a modest increase in predictivity relative to the baseline model with random weights; and (3) L2 adversarial training induces significantly more alignment than standard supervised training in both V1 and V2 (Parthasarathy et al., 2024). Finally, we see that our layerwise-trained ST-MMCR model performs similarly to the robust (adversarially trained) baseline and outperforms end-to-end supervised learning in all three areas.

We additionally ablate two key aspects of our training algorithm: the stage-wise isolation of gradients, and the application of losses at internal layers of the network. To do so, we trained one network with an identical loss function and architecture, but without "gradient cutting," so for example $\mathcal{L}_{MMCR}(\mathbf{C}^{(2)})$ impacts the parameters of the first stage $\theta_1$; we denote this ablation "E2E Gradient" in Fig. 8. We additionally train a third model that uses end-to-end gradients, but further ablate all internal losses, so $\mathcal{L}_{MMCR}(\mathbf{C}^{(3)})$ alone is used to train all three stages ("E2E Loss" in Fig. 8). Both modifications produce modest reductions in neural predictivities relative to our default

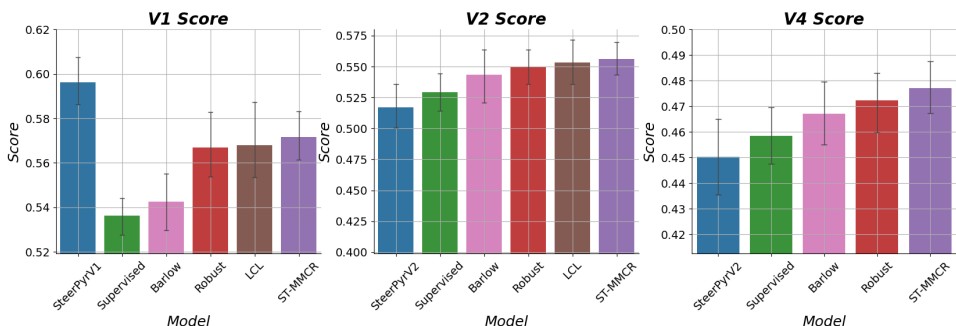

Figure 3: **Neural predictivity across baseline models.** For each model and neural dataset, we apply the procedure described in 2.4 for all model layers and report the highest score. For each panel, the lowest score on the ordinate corresopnds to the mean performance of an architecture-matched model with random weights. Error bars indicate 95% confidence intervals over train-test splits.

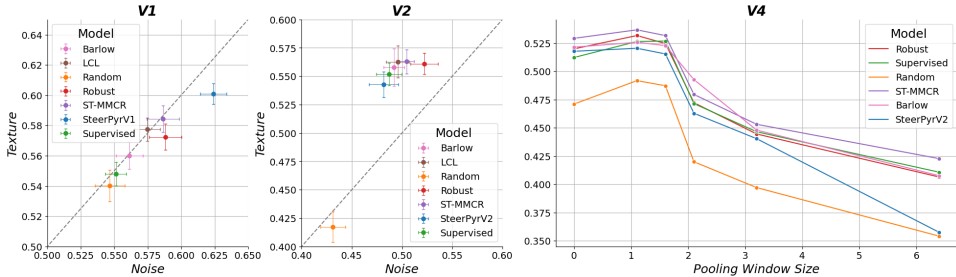

Figure 4: **Neural predictivity by stimulus type.** Model predictions for each neural dataset separated by stimulus type. In V1 and V2 (left two plots) images are split into naturalistic textures and spectrally matched noise images. In V4, images are parameterized by the size of the windows over which they are "texturized."

settings across all three brain areas, suggesting at the very least that our simplified credit assignment procedure is not harming model-brain alignment (see Appendix A.1 Fig. 8 for full results).

## 3.1 PARTITIONING PREDICTIVITY

As demonstrated in Fig. 3, and noted in previous literature (Cadena et al., 2024; Conwell et al., 2022), models obtained using the same architecture and training dataset tend to have similar predictivity of cortical responses.This does not necessarily imply that all such models are capturing the same aspects of neural responses, or encoding those aspects in the same format as the biological neurons (since predictions are based on linear regression from a large number of model neurons). To address this, we examined two means of "decomposing" each model's neural predictivity.

**Partitioning by stimulus parameters.** Each neural dataset under consideration leverages images that are parametrically generated or distorted. Freeman et al. (2013) presented synthesized naturalistic textures and spectrally matched noise images (having identical 2nd-order spectral energy but without higher-order structure). Lieber et al. (2024) generate a continuum ranging from natural images to their fully texturized counterparts by drawing samples that are statistically matched over progressively larger "pooling windows" (a pooling window size of 0 corresponds to an unperturbed natural image and the largest window size corresponds to a fully texturized counterpart – see A.1 for examples).

Figure 4 shows comparisons of neural predictivity for texture and noise images in V1 and V2, and across pooling window sizes in V4. For V2 units (but not V1 units) responses to naturalistic textures are significantly better predicted than their spectrally matched noise counterparts (left two panels, Fig. 4, and for V4 units predictivity falls sharply for pooling window sizes greater than 1.6 visual degrees. These behaviors are generally shared between candidate models.

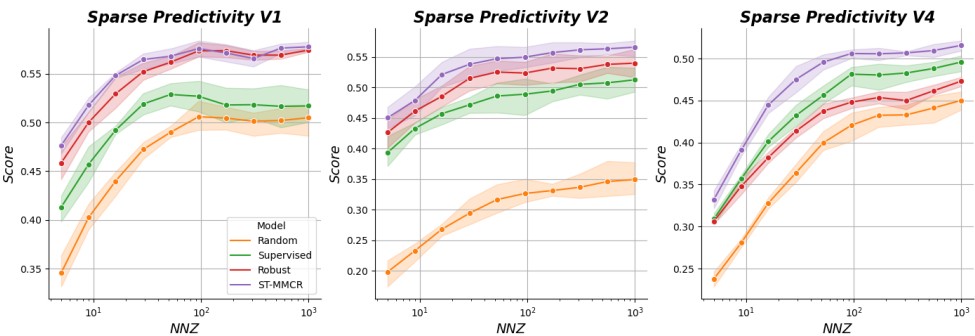

Figure 5: **Sparse neural predictivity.** For each model-benchmark pair we use lasso to select $k$ model responses to predict each individual neuron's firing rate via ridge regression. We vary $k$ over 10 logarithmically spaced values between 5 and $1,000$.

**Partitioning by number of model units, using sparse regression.** Even in the case where two models make identical predictions for neural responses to *all images*, they may encode the information used to make these predictions in disparate formats. As an extreme example, consider an activation map produced by a candidate model $a \in \mathbb{R}^{c \times h \times w}$ where only a single element is predictive of an individual neuron's firing rate. In such a case the optimal linear mapping uses a weight vector that is one-hot; an alternative model might produce a mapping with all non-zero coefficients, yet the composition of network backbones with their respective nonlinear maps can compute the same function. To assess the extent to which linearly predictive information is isolated or distributed across our suite of candidate representations, we trained linear mappings that are restricted to use $k$ non-zero coefficients. Specifically, we perform feature selection on the training set of each split by adjusting the strength of an $L1$ (LASSO) regularizer until a sparse set of $k$ coefficients is selected, then re-fit neural responses with the selected features using ridge regression (i.e., the "relaxed LASSO" – see Appendix A.5 for details).

The results of this analysis are shown in Fig. 5. First we note that trained models are generally better separated from their random-weight counterparts in the highly sparse regime. For example in V1 (left panel), the random model only narrows the gap to the supervised model when it has access to hundreds of model units to interpolate individual neural firing rates. In V2 (center panel) models are primarily separated by the amount of information offered by the few most predictive units, with the predictivity gains per unit (slopes of each curve) being approximately matched. These results suggest that bottlenecked regressions can provide a useful signal, particularly for disentangling alignment induced by training from alignment arising due to the architecture.

## 4 BEHAVIORAL ALIGNMENT

Several studies have provided evidence that representations whose early stages are more aligned with primary visual cortex support more human-like behavior in object representation tasks (Dapello et al., 2020; Marques et al., 2021; Feather et al., 2023). Recently, Parthasarathy et al. (2024) extended this result by training a downstream classifier on a self-supervised V2 front end model. However, it remains unclear whether this trend will hold in deeper cortical areas, or even whether V1 or V2 predictivity is more important for behavioral alignment.

We explicitly investigate this question by training classifiers to operate on responses of models of each brain-area in Section A.4 with results shown in Fig. 10. We find that the classifier trained on top of the V2-like layer has the highest out-of-distribution generalization and alignment to human behavior, and we thus compare its performance with our baseline models in Fig. 6. Specifically, we utilized the suite of out-of-distribution (OOD) stimuli introduced by Geirhos et al. (2021) to test the models' ability to generalize as well as their propensity to make category judgments similar to human observers. The ST-MMCR model exhibits substantially lower accuracy on the standard ImageNet recognition task (Fig. 6 left panel) than the fully supervised model. But the ST-MMCR with a V2 frontend substantially outperforms *all* architecture-matched baselines in terms of OOD generalization, human choice, and error consistency (Fig. 6 right three panels).

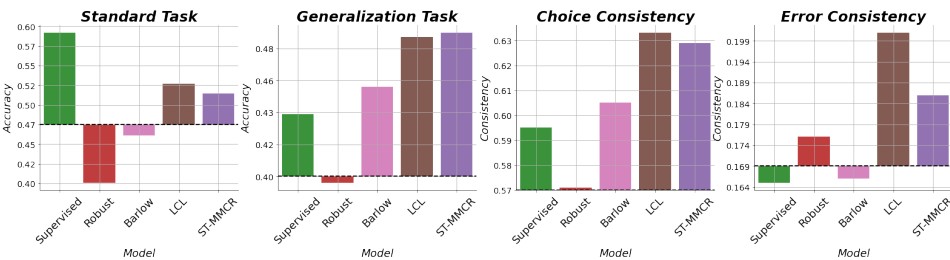

Figure 6: **Performance and alignment with human classification.** In the first two panels we show the top-1 accuracy on the standard ImageNet-1k validation set and on the out-of-distribution images from Geirhos et al. (2021), respectively. In the right two panels we show the rate at which models make the same choice as human observers, and the alignment between model and human error patterns (as described in Geirhos et al. (2021)). In all panels, the black dashed lines indicates the performance of a classifier trained on top of a front end with random weights, with the same number of frozen layers as the ST-MMCR model.

## 5 DISCUSSION

We have developed a novel local layerwise self-supervised learning scheme, termed ST-MMCR, which naturally implements "complexity matching" by leveraging dilating spatial and temporal receptive fields, rather than relying on hand-crafted augmentation schemes as was done in previous work (Parthasarathy et al., 2024). Our formulation is appealing from the perspective of biological plausibility. First, backpropagation of error signals arising from a single global objective is widely considered implausible as a model for biological brains (Stork, 1989), and the stagewise local computation of our model offers a potential alternative. Moreover, many converging lines of evidence indicate that primate neocortex is constructed of local circuits that are replicated - to first approximation the entire cortical sheet consists of identical computational units (Douglas & Martin, 2004). Specific computations arise through parallel, sequential and recurrent connectivity of these circuits, with details largely learned during development and in response to life experience. This provides motivation for convolutional neural networks in general, and for a local common objective function computation. Both the architecture and objective of our model aim to satisfy this description: the three stages are implemented with identical computations, apart from the change in temporal pooling duration. Rectified linear operations, normalization of responses over groups of "cells", and spatial and temporal averaging are common elements of many cortical models (Carandini & Heeger, 2012).

To better understand how candidate models differ beyond overall predictivity, we performed a series of fine-grained analyses. By partitioning neural data according to stimulus parameters, we revealed systematic differences in which response components are predictable from model activations. Notably, V2 representations showed stronger alignment with naturalistic texture responses, while V4 predictivity degraded as natural images were texturized. Sparse regression analyses revealed that models initialized with random weights can achieve modest predictivity by interpolating neural responses using a large number of units, whereas trained models achieve comparable or superior predictivity using a much smaller set of informative features. These results suggest that bottlenecked regressions can surface alignment signals that are otherwise masked by overparameterization. Finally we found that our biologically motivated pretraining procedure induced increased alignment to human behavior in object classification tasks. Taken together, these results suggest that simplified credit assignment procedures are not only capable of producing brain-like representations, but the biological constraints they employ may play a key role in shaping the structure of real visual systems.

Future work can improve upon our approach by increasing the ecological relevance of the training stimuli, i.e. by using a suitably large and diverse set of natural videos. Furthermore, the final MMCR loss, which relies on nuclear norm computation, does not currently have an obvious counterpart in the world of biological modeling, but dynamic circuits for optimizing other objectives may offer a path for future development Pehlevan et al. (2017). Finally it would be interesting to consider alternative objectives that incorporate either more nuanced features of capacity maximization (Yerxa et al., 2024; Chou et al., 2024) or more general non-ellipsoidal manifold geometries.

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

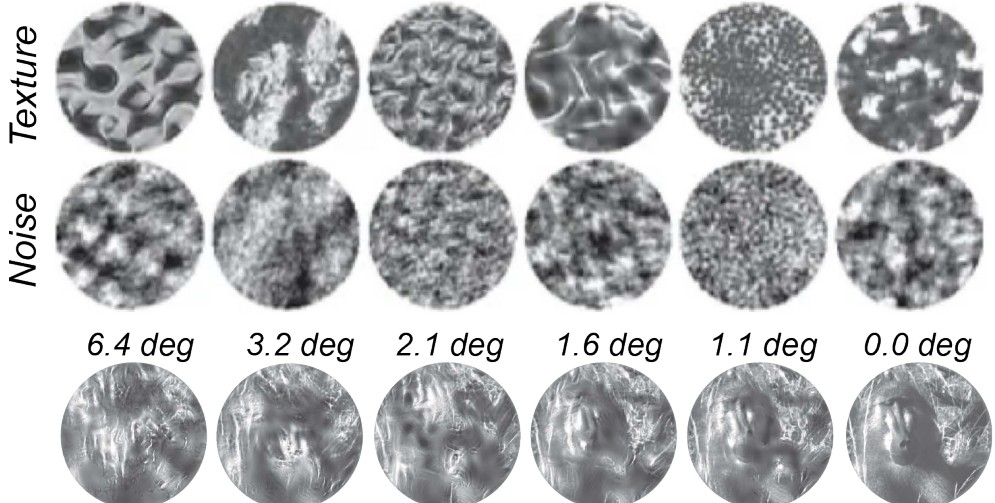

Figure 7: **Example stimuli used for each of the three neural datasets.** The top two rows show images used in the V1/V2 experiments of Freeman et al. (2013) and the last row shows images used in the V4 experiment of Lieber et al. (2024). These figures are reproduced from the original works with the permission of the authors.

Richard Zhang. Making convolutional networks shift-invariant again. In *International conference on machine learning*, pp. 7324–7334. PMLR, 2019.

Chengxu Zhuang, Siming Yan, Aran Nayebi, Martin Schrimpf, Michael C Frank, James J DiCarlo, and Daniel LK Yamins. Unsupervised neural network models of the ventral visual stream. *Proceedings of the National Academy of Sciences*, 118(3):e2014196118, 2021.

Corey M Ziemba, Jeremy Freeman, J Anthony Movshon, and Eero P Simoncelli. Selectivity and tolerance for visual texture in macaque v2. *Proceedings of the National Academy of Sciences*, 113(22):E3140–E3149, 2016.

# A  APPENDIX

## A.1  NEURAL DATASET AND EVALUATION DETAILS

We used the neural dataset from Freeman et al. (2013); Ziemba et al. (2016) for evaluating alignment to areas V1 and V2, and data from Lieber et al. (2024) for evaluating alignment to V4. Below we describe the relevant aspects of the stimuli used in these experiments, and provide further details regarding the evaluation protocol.

**Stimulus generation.**   Stimulus generation for the V1 and V2 datasets begins with a set of 15 photographs of natural visual textures (i.e. tree bark). For each texture image, a set of synthetic images is generated by initializing to samples of white noise and and updating pixel values via gradient descent to match a set of statistics defined by the parametric texture model of Portilla & Simoncelli (2000). Matched noise stimuli are generated by taking the Fourier transform of each texture image and scrambling the resulting phases while preserving the amplitudes, producing images with matched spectral energy but lacking the higher order structures characteristic of the texture images. Example synthetic textures and their spectrally-matched counterparts are shown in Fig. 7.

The images used in the V4 dataset are derived from the UPenn Natural Image Database (Tkačik et al., 2011). These base images are then "texturized," using an analogous statistical matching procedure to the one described above, but modified so that the texture statistics are matched within overlapping pooling regions of varying sizes. Because the images were presented at $6.4°$ of visual angle, when the pooling region is this size, this corresponds to the texture synthesis procedure used

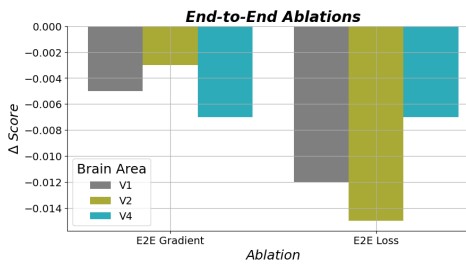

Figure 8: **Impact of localized gradients and intermediate loss functions.** We train two models without layerwise gradient computations (left group of columns), and additionally without the application of losses at intermediate stages (right group of columns), and evaluate each for neural predictivity. We report the change in score relative to the default settings (the ST-MCMCR columns in Fig. 3.

for V1/V2. Conversely a pooling region size of $0.0°$ corresponds to matching each individual pixel value, producing an image identical to the original base image. Example stimuli for each pooling region size are shown in the bottom row of Fig 7.

**Evaluation protocol.** We follow the procedure described in Section 2.4, using 10 cross validation folds for the V1/V2 datasets and 5 for the V4 dataset. Here we describe the preprocessing steps applied to the raw stimuli used in each set of stimuli used to obtain the model responses. One key parameter is the assumed field of view input resolution associated with models trained on images with arbitrary resolution. For the V1/V2 stimuli and each trained baseline model (the standard and adversarially trained AlexNets) as well as the vast majority of models trained on ImageNet, an input size of 224 pixels and $8°$ field of view are known to be optimal in terms of predictivity (Schrimpf et al., 2018), and so we simply adopt this convention for our model. Because in the V1/V2 experiment stimuli were displayed at $4°$ of visual angle, the stimuli are resized to the appropriate resolution, and placed in the central $4°$ of the input image using a circular aperture with a 112 pixel diameter. For the V4 dataset the same procedure is applied, with a modification made to account for the fact that these stimuli were presented at a $6.4°$ field of view. Additionally we apply the standard normalization transform that was employed while models are training on ImageNet. Finally it is worth noting that, following Parthasarathy et al. (2024) we skip all preprocessing steps when extracting responses from pyramid-based models.

**End-to-End ablation.** We ablated the "layerwise" aspect of our training procedure in two steps by training networks with (1) identical loss functions and architecture, but without "gradient cutting," and (2) full end-to-end gradients, but and without any internal losses. The results are shown below in Fig. 8.

### A.2 INPUT TRANSFORMATION DETAILS

The complete set of image transformations used to obtain frames for training the ST-MMCR representation is given in Table 1. To obtain the first and final frames, we apply two random resized cropping operations. Intermediate frames are then obtained by linearly interpolating the coordinates of the anchor frames, an example sequence of frames is shown in Fig. 9. Each frame is individually subjected to a series of photometric distortions: modest contrast and brightness modulation, a stochastic between color and grayscale, and a chance for the addition of gaussian noise with standard deviation randomly selected between $0.04$ and $0.1$.

For the lower complexity transformations of Fig. 2, we only change the spatial transformation step. To reduce the input feature complexity, we first resize images so the shortest edge is 224 pixels, then take center crops of sizes $[33, 56, 112]$ for complexities 1, 2, and 3, respectively. To modulate the transformation complexity we then set the minimum scale of the random resized cropping operations for each complexity to $0.9$, $0.6$, and $0.3$, which are applied to the center cropped image patches. The fourth complexity transformation is the default for ST-MMCR described above. Thus across the x-axis of Fig. 2, inputs have more complicated visual features as the fraction of pixels included in the synthetic videos grows, and the extent of the transformations grows as the anchor crops are allowed to be separated by larger distances in pixel space.

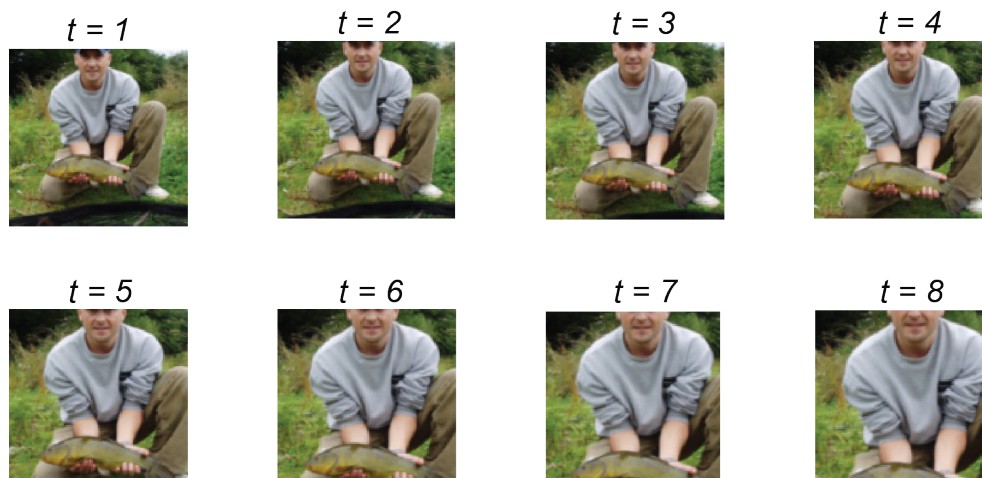

Figure 9: **Example synthetic video frames.** The first and final frames are obtained by taking two random crops of a single image, and intermediate frames are obtained by linearly interpolating crop parameters. Finally each frame is resized. This transformation is a crude way of simulating the smooth motion that can occur as the relative position of the observer and objects in the scene change over time.

| Parameter | $t = 0$ | All $t$ | $t = T$ |
|---|---|---|---|
| Minimum random crop scale | 0.08 | | 0.08 |
| Random crop resize output size | 224 | | 224 |
| Color jittering probability | | 0.8 | |
| Color dropping probability | | 0.2 | |
| Brightness adjustment max | | 0.2 | |
| Contrast adjustment max | | 0.2 | |
| Gaussian noise probability | | 0.5 | |

Table 1: Parameters for the default transformation scheme used to train the ST-MMCR model.

### A.3 ARCHITECTURE AND OPTIMIZATION DETAILS

As described in Section 2.2, we use the first and second convolutional layers of the AlexNet architecture to parameterize the first and second stages of the ST-MMCR representation, and the third and fourth convolutional layers for the third stage. For completeness we give a full description of each operation in Table 2. Because computing our objective involves taking small crops of internal feature maps, we precede MaxPooling operations with a fixed (not trained) blurring operation to mitigate the impact of aliasing (using the Blurred max pooling operation from Zhang (2019)). Finally, we note that our implementation includes a batch normalization that precedes each rectifier, and uses a third pooling operation that appears "one stage early" relative to the original AlexNet architecture. While these represent a small departure from the more standard choices present in the supervised and robust baseline models, we observed that the absence of the batch normalization in these models did not significantly impact their neural predictivity or behavioral alignment scores (a result previously reported in Parthasarathy et al. (2024)). We additionally provide exact details for each stages associated projector network in Table 3.

The parameter optimization was performed using the Adam optimizer (Deng et al., 2009; Kingma & Ba, 2014), and iterating for 10 epochs of the ImageNet-1k dataset with a batch size of 32 and a constant learning rate of $1e - 3$.

| $f_{\theta_1}$ | **Conv2d**(`in_channels=3, out_channels=64, ks=11,`
`stride=4, padding=5, padding_mode='reflect'`)
**BatchNorm2d**(64)
**ReLU**()
**BlurMaxPool2d**(`ks=3, stride=2, padding=1`) |
|---|---|
| $f_{\theta_1}$ | **Conv2d**(`in_channels=64, out_channels=192, ks=5,`
`stride=1, padding=2, padding_mode='reflect'`)
**BatchNorm2d**(192)
**ReLU**()
**BlurMaxPool2d**(`ks=3, stride=2, padding=1`) |
| $f_{\theta_3}$ | **Conv2d**(`in_channels=192, out_channels=384, ks=3,`
`stride=1, padding=1, padding_mode='reflect'`)
**BatchNorm2d**(384)
**ReLU**()
**Conv2d**(`in_channels=384, out_channels=256, ks=3,`
`stride=1, padding=1, padding_mode='reflect'`)
**BatchNorm2d**(256)
**ReLU**()
**BlurMaxPool2d**(`ks=3, stride=2, padding=1`) |

Table 2: Detailed description of the architecture for each stage of the ST-MMCR representation.

| $g_{\phi_1}$ | **Conv2d**(`in_channels=64, out_channels=64, ks=1,`
`stride=1, padding=0`)
**BatchNorm2d**(64)
**ReLU**()
**Conv2d**(`in_channels=64, out_channels=2048, ks=1,`
`stride=1, padding=0`) |
|---|---|
| $g_{\phi_2}$ | **Conv2d**(`in_channels=192, out_channels=192, ks=1,`
`stride=1, padding=0`)
**BatchNorm2d**(192)
**ReLU**()
**Conv2d**(`in_channels=192, out_channels=2048, ks=1,`
`stride=1, padding=0`) |
| $g_{\phi_3}$ | **Conv2d**(`in_channels=256, out_channels=256, ks=1,`
`stride=1, padding=0`)
**BatchNorm2d**(256)
**ReLU**()
**Conv2d**(`in_channels=256, out_channels=2048, ks=1,`
`stride=1, padding=0`) |

Table 3: Detailed description of the architecture of the projector network associated with each stage of the ST-MMCR representation.

## A.4 BEHAVIORAL EVALUATION DETAILS

**Front-End depth experiment.** As a step towards determining to which brain area alignment is most critical for inducing human-like behavior on classification tasks we trained ImageNet-1k classifiers on top of V1/V2/V4 front-ends. Specifically, we froze our pretrained model at the first, second or third stage, and appended the remaining stages of the standard AlexNet architecture to each front end (see below for details on the classifier training procedure). In this experimental design, the V2 based model is handicapped relative to the V1 (and the V4 relative to the V2) model as it has fewer trainable parameters dedicated to the classification task. To control for this we additionally trained classifiers on top of randomly-initialized front-ends that were frozen at matched points along the hierarchy to isolate the impact of ST-MMCR pretraining.

We measure each models in-distribution generalization (performance on the Standard ImageNet-1k validation set), as well as out-of-distribution generalization and alignment with human behavioral

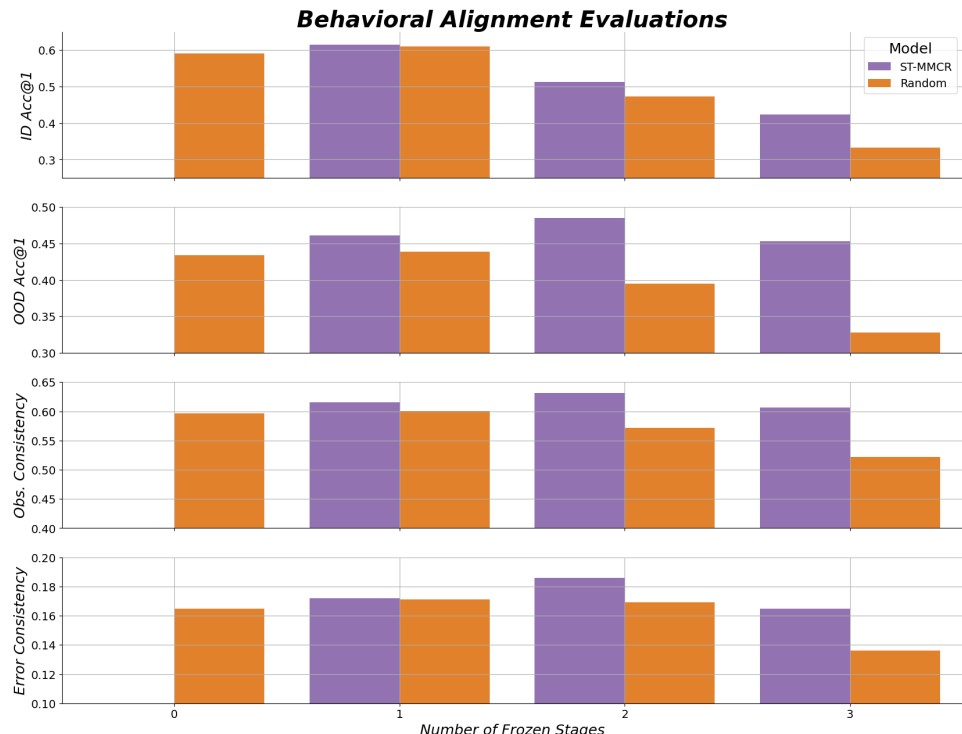

Figure 10: **Behavioral evaluation of V1, V2, and V4 front-ends.** All networks are evaluated on the ImageNet-1k task (top row), OOD generalization on the 20 datasets from Geirhos et al. (2021) (second row), and the rate of agreement between model and human category choices on said datasets (bottom two rows).

choices on the suite of OOD datasets described in Geirhos et al. (2021). The results are summarized in Fig. 6. Despite marked decreases in in-distribution performance when freezing multiple stages (top row), OOD accuracy, observational consistency, and error consistency (as defined in Geirhos et al. (2021) and re-described in A.4) remain similar or superior to the standard supervised AlexNet.

**Classifier training procedure.** We trained classifiers on top of each of the three stages of the ST-MMCR architecture by "completing" the AlexNet architecture (with batch normalization) using randomly initialized modules. For example, when training on top of the outputs of $f_{\theta_1}$, the second convolutional stage was freshly initialized to have random trainable parameters, while the parameters of the first convolution were frozen to the ST-MMCR values (i.e., this corresponds to 1 frozen stage, as indicated on the x-axis of Fig. 6). Note that for the outputs of the third stage of ST-MMCR we omitted the final downsampling operation to maintain consistency across classifier architectures (see Appendix A.3 for justificaiton). As a result of this strategy, the V2-frontend classifier has fewer trainable parameters than its V1-frontend counterpart. To better isolate the impact of ST-MMCR pretraining (rather than differences in model capacity), we compare each model to a matched version in which the frontend is randomly initialized and frozen. This pairing places the pretrained and randomly initialized models on more equal footing across conditions with varying numbers of frozen stages.

The trainable parameters of each classifier were trained on the ImageNet-1k dataset using the standard cross entropy loss and stochastic gradient descent with momentum and weight decay for 100 epochs. We used a batch size of 512, an base learning rate of 0.1 which decayed by a factor of 10 every 30 epochs, a momentum value of 0.9, and the weight decay penalty set to $1e - 4$.

**Evaluation metrics and additional results.** In addition to reporting the standard accuracy of each model on the ImageNet-1k validation set, we evaluated the behavioral alignment of models using the datasets from Geirhos et al. (2021) (using the official implementation which can be found

 The data consists of human categorization responses to 17 out-of-distribution (from the model's perspective) tasks. These tasks are generated by distorting photographic images by applying various style-transfer techniques or through the application of parametric degradations such as the addition of noise or blurring and were originally described in Geirhos et al. (2018a;b).

In addition to reporting the average accuracy on this suite of OOD tasks, we show the observation and error consistencies between each model and human responses. The observation consistency measures the rate at which the model and human both categorize a given sample either correctly or incorrectly, and the error consistency measures whether the choice consistency is higher or lower than what would be obtained from a pair of random models with accuracies matched to the model and human respectively. For complete descriptions see Geirhos et al. (2021).

In addition to the results for the ST-MMCR and randomly initialized front-end models visualized in Fig. 6, we show results for each of these metrics in Tables 4 and 5.

| model | accuracy diff. ↓ | obs. consistency ↑ | error consistency ↑ | mean rank ↓ |
|---|---|---|---|---|
| ST-MMCR-V2 | **0.095** | **0.631** | **0.186** | **1.000** |
| ST-MMCR-V1 | 0.113 | 0.615 | 0.172 | 3.000 |
| ST-MMCR-V4 | 0.112 | 0.607 | 0.165 | 3.667 |
| RI-V1 | 0.123 | 0.601 | 0.173 | 4.000 |
| Robust | 0.145 | 0.573 | 0.176 | 4.667 |
| Supervised | 0.118 | 0.597 | 0.165 | 5.333 |
| RI-V2 | 0.147 | 0.572 | 0.169 | 6.333 |
| RI-V4 | 0.189 | 0.522 | 0.136 | 8.000 |

Table 4: **Behavioral alignment metrics for all models and baselines considered in this work.** The first column shows the difference in OOD accuracy between a model and human observer, and subsequent columns are described in the text. The -V1 indicates that one frozen stage served as a front-end for a classifier, and -V2 2 stages and -V4 three stages. Finally the RI- models correspond to randomly initialized frontends.

| model | OOD accuracy ↑ | rank ↓ |
|---|---|---|
| ST-MMCR-V2 | **0.485** | **1.000** |
| ST-MMCR-V1 | 0.461 | 2.000 |
| ST-MMCR-V4 | 0.453 | 3.000 |
| RI-V1 | 0.439 | 4.000 |
| supervised | 0.434 | 5.000 |
| RI-V2 | 0.395 | 6.000 |
| robust | 0.391 | 7.000 |
| RI-V4 | 0.328 | 8.000 |

Table 5: **Average accuracy on the OOD tasks from Geirhos et al. (2021).** The model naming convention is the same as described in the caption of Table 4.

### A.5  SPARSE REGRESSION DETAILS

To investigate how predictivity varies as a function of the number of model units used to interpolate a neural response, we used a variant of the relaxed LASSO procedure Meinshausen (2007). Concretely, for each neural unit and train test split we first compute the LASSO regularization path on the train data to obtain estimators with varying levels of sparsity (Friedman et al., 2010). We used `sklearn`'s implementation of lasso path, and find 200 different estimators using regularization coefficients that varied over 8 orders of magnitude. After this step, we selected the mappings that had nearest the desired levels of sparsity, $k$, which we set to be 10 logarithmically spaced values between 5 and 1000. Next we take the selected features (i.e. the approximately $k$ features with non-zero coefficients), and refit the training data using only this reduced feature space, via ridge regression. We

repeat this process for each neuron and each train-test split, and report the final score by taking the median over neurons and mean over splits, as described in Section 2.4.

## A.6 A Circuit for Maximizing Nuclear Norm

Consider the following objective function:

$$\min_{\mathbf{Y} \in \mathbb{R}^{n \times T}} \|\mathbf{X} - \mathbf{Y}\|_F^2 + \lambda \|\mathbf{Y}\|_F^2 - 2\lambda \|\mathbf{Y}\|_*,$$

First, we note that the nuclear norm of $\mathbf{Y}$ can be written as the solution to an optimization problem:

$$\|\mathbf{Y}\|_* = \max_{\mathbf{Z} \in \mathbb{R}^{n \times T}} \operatorname{Tr}(\mathbf{Y}^\top \mathbf{Z}) \quad \text{subject to} \quad \frac{1}{T}\mathbf{Z}\mathbf{Z}^\top = \mathbf{I}.$$

This optimization problem can be rewritten using the method of Lagrange multipliers:

$$-2\lambda \|\mathbf{Y}\|_* = \max_{\mathbf{Z} \in \mathbb{R}^{n \times T}} \min_{\mathbf{M} \in \mathbb{R}^{n \times n}} \left( -2\lambda \operatorname{Tr}(\mathbf{Y}^\top \mathbf{Z}) + \operatorname{Tr}\left[\mathbf{M}\left(\frac{1}{T}\mathbf{Z}\mathbf{Z}^\top - \mathbf{I}\right)\right] \right).$$

Substituting this term and exchanging the order of optimization in the original objective yields,

$$\max_{\mathbf{M} \in \mathbb{R}^{n \times n}} \min_{\mathbf{Y} \in \mathbb{R}^{n \times T}} \min_{\mathbf{Z} \in \mathbb{R}^{n \times T}} \left\{ \|\mathbf{X} - \mathbf{Y}\|_F^2 + \lambda \|\mathbf{Y}\|_F^2 - 2\lambda \operatorname{Tr}(\mathbf{Y}^\top \mathbf{Z}) + \operatorname{Tr}\left[\mathbf{M}\left(\frac{1}{T}\mathbf{Z}\mathbf{Z}^\top - \mathbf{I}\right)\right] \right\}$$

Gradient steps on our optimization variables are,

$$\Delta \mathbf{Y} = \gamma\left(\mathbf{X} + \lambda \mathbf{Z} - (1+\lambda)\mathbf{Y}\right),$$
$$\Delta \mathbf{Z} = \gamma\left(\mathbf{Y} - \mathbf{M}\mathbf{Z}\right),$$
$$\Delta \mathbf{M} = \eta\left(\frac{1}{T}\mathbf{Z}\mathbf{Z}^\top - \mathbf{I}\right)$$

Inspecting these dynamics, we can see this corresponds to a neural circuit with a population of primary neurons $\mathbf{Y}$ that receives feedforward drive from the inputs, feedback excitation from a population of interneurons $\mathbf{Z}$, and recurrently inhibits itself. The interneuron population receives feedforward drive from the primary population and recurrent inhibition from itself via the synaptic weight matrix $\mathbf{M}$, and the synaptic update steps for $\mathbf{M}$ constitute a local learning rule.

