# OpenReview forum: "Modeling Visual Cortex by Maximizing Layerwise Multiscale Manifold Capacity"
_ICLR.cc/2026/Conference — Submitted to ICLR 2026_

### Official Review · Reviewer_QTgA · 2025-10-23

**Soundness:** 3
**Presentation:** 4
**Contribution:** 3
**Rating:** 6
**Confidence:** 4

**Summary:**

This paper proposes ST-MMCR, a layerwise self-supervised learning approach for predicting neural data from the ventral visual stream of primates and human psychophysical data on an object classification task. The key innovation, building on top of the manifold capacity framework, is using spatiotemporal pooling regions that scale with receptive field size to implement "complexity matching" - ensuring task difficulty aligns with computational capacity at each stage. The approach trains on synthetic videos generated from ImageNet images and evaluates on neural recordings from macaque V1, V2, and V4, as well as human behavioral data. Results show competitive or superior performance compared to supervised and adversarially-trained baselines.

**Strengths:**

## Overall Assesment

This is a strong and well-executed submission that will be of significant interest to the ICLR community, particularly researchers working at the intersection of neuroscience and machine learning. The paper makes meaningful contributions to biologically-plausible visual representation learning while achieving competitive empirical results. I believe the strengths substantially outweigh the weaknesses, though several points warrant clarification during the discussion phase (detailed below).


## Strengths

1. **Biologically-motivated architecture and learning scheme.** The layerwise learning approach with gradient isolation directly addresses a well-known implausibility of backpropagation in biological systems. Importantly, this work demonstrates that such biologically-constrained learning can achieve state-of-the-art neural predictivity, challenging the notion that biological plausibility requires sacrificing performance. The use of dilating spatiotemporal pooling regions is particularly elegant, directly motivated by the expanding receptive field sizes observed across the ventral visual stream.

2. **Well-designed synthetic video approach.** The decision to use synthetic videos generated from ImageNet images cleverly balances ecological validity with experimental control. This avoids confounding distribution shifts that would arise from switching to real video datasets while allowing precise parametric control over spatiotemporal transformations. While I believe real video validation would further strengthen the work (see weaknesses), this represents a reasonable and methodologically sound first step.

3. **Comprehensive neural evaluation.** The evaluation framework is thorough and multi-faceted: (a) three cortical areas spanning early-to-mid visual hierarchy (V1, V2, V4), (b) stimulus partitioning analyses revealing which response components are captured, (c) sparse regression analysis examining representational format, and (d) behavioral alignment on out-of-distribution tasks. This goes well beyond standard aggregate predictivity metrics.

4. **Competitive performance with biological constraints.** Achieving performance comparable to or exceeding adversarially-trained models—while using simpler, more biologically plausible learning rules—is a significant achievement. This demonstrates that biological constraints need not be a handicap and may actually provide useful inductive biases.

5. **Analysis of model-brain alignment.** The sparse regression analysis (Fig. 5) and stimulus partitioning (Fig. 4) provide important insights into how models achieve predictivity, not just how much. This reveals that models with similar aggregate scores can differ substantially in their representational structure. Such analysis is crucial for moving beyond superficial alignment metrics and I believe it's complete and serves the paper's goal.

6. **Clear presentation.** The paper is well-written with effective visualizations, particularly Figure 1, which immediately conveys the core approach. The organization facilitates understanding of both the technical contributions and their biological motivation.

**Weaknesses:**

## Weaknesses

1. **Biological plausibility of MMCR loss.** The paper emphasizes biological plausibility but acknowledges (pg. 9) that "the final MMCR loss, which relies on nuclear norm computation, does not currently have an obvious counterpart in the world of biological modeling." This is a significant limitation for the central framing. The computation requires: (a) global normalization across all centroids, (b) SVD computation, which are both implausible. More discussion of how this might be implemented biologically is needed.

2. **Limited architectural generalization.** Using only AlexNet limits the impact. While the authors justify this choice, modern architectures (ResNets, Vision Transformers) would strengthen claims about the general utility of the approach. It's unclear if the spatiotemporal pooling strategy would work as well with skip connections or attention mechanisms, that are part of standard (supervised) sota models in the [brainscore](https://www.brain-score.org/vision/leaderboard/) leaderboard (M. Schrimpf et al, BioArxiv 2018).

Minor Weaknesses

1. **Incomplete comparison to layerwise learning methods.** The paper mentions CLAPP and LPL but doesn't directly compare to them experimentally. Given that layerwise learning is central to the contribution, head-to-head comparison on neural predictivity would be valuable. The end-to-end ablations (Fig. 8) are useful but don't substitute for comparison to other layerwise approaches.

2. **Limited ablation studies** Key design choices lack thorough ablation: Why 8 frames specifically? How sensitive are results to this choice? How important is the specific schedule of pooling sizes (sf, sf/2, sf/4)? What about different video transformation strategies?

3. **Hyperparameter sensitivity** The paper doesn't discuss sensitivity to key hyperparameters (learning rate, batch size, projection head architecture). This makes reproducibility and extension challenging.

**Questions:**

I want to emphasize that I am on the high end of 6 and would rate this a 7 if that option were available. The work has substantial merit and clear acceptance-worthy contributions. I am enthusiastic about this paper and would be happy to raise my score to 8 following the discussion period if the authors can satisfactorily address the weaknesses above and questions below.

## Questions

1. Can you provide intuition or preliminary ideas for how the nuclear norm computation might be approximated with biologically plausible operations?

2. How does computational cost compare to supervised training and adversarial training? Is there a trade-off between biological plausibility and computational efficiency?

3. Have you considered applying this approach to modern architectures (ResNets, Transformers)? Even a preliminary result would strengthen generalization claims.

4. What happens with longer videos (16, 32 frames)? Does performance continue to improve or saturate?

5. V2 frontend superiority (Fig. 10, Appendix A.4): The finding that classifiers trained on V2-like representations outperform both V1 and V4 frontends on behavioral tasks is intriguing but under-explored. This is counter-intuitive since V2 has fewer trainable parameters in the classifier (more frozen stages) and V4 representations are deeper/more abstract. Can you provide mechanistic insight into why V2 is the "sweet spot"? Is this related to: (a) the level of invariance/selectivity, (b) superior texture representations (as suggested by Fig. 4), (c) manifold geometry properties, or (d) something specific about how ST-MMCR structures V2 representations? This could be a key finding that warrants more prominence and discussion in the main text.

---

> ### Author Response · Authors · 2025-11-23
> **Response to Reviewer QTgA (Part 1)**
>
> Thank you for your thorough and encouraging review of our work. Below we respond point by point to each weakness and question:
>
> - **W1**: We agree that this is a significant limitation of our work in its current form. We are actively pursuing the development of a circuit that could plausibly implement a similar optimization algorithm. Please see the detailed response to a similar question from Reviewer naJ4 (W6) (and the new Appendix A.6 where we describe some initial progress in this direction).
>
> - **W2**: We agree that this is a weakness of the current work. While we will not be able to conduct sufficient experiments to provide definitive evidence on this front during the rebuttal, we can share some preliminary results from the initial phases of the development of this project. We had experimented with using the ST-MMCR loss as a regularizer acting in addition to an end-to-end supervised learning signal by applying the ST-MMCR loss to internal layers during training.
>
>   In this setting we were using a ResNet-50 architecture and applying the ST-MMCR loss to the outputs of each of the four residual blocks. In this setting we found that the impact of our ST-MMCR loss was significantly more pronounced in earlier layers, which played a role in leading us to exclusively focusing on ST-MMCR loss as a stand alone layerwise objective in a shallower network. We would be happy to share more details and the preliminary results from these experiments should they be helpful for your evaluation, but opted not to include them as the considered setting is substantially different from the one in the paper.
>
>   While this does not directly speak to the applicability of our framework to deeper networks in a 1-to-1 sense, and we suspect there would be significant challenges with finding an appropriate way to scale complexity in less “vanilla” architectures (i.e. with skip connections and other features “muddying the waters” of defining the expressivity of each stage), it is an encouraging datapoint in this direction.
>
> - **W3**: We wholeheartedly agree with this sentiment. Since the initial submission we have implemented the layerwise approach of LCL [1] and have included a direct comparison in the figures of the updated manuscript. Based on the results of that paper, LCL is the “strongest” layerwise approach to compare to, but we are in the process of implementing LPL and CLAPP using matched architectures and training datasets as well and would aim to include direct comparisons in the final version of this paper should it be accepted.
>
> - **W4**: Coarse experiments while developing this work showed that the results were insensitive to increasing the number of frames beyond the minimum 8, so we made this choice in the name of efficiency. We would be happy to add an explicit ablation, but time and compute restrictions prevent this from being carried out during the rebuttal.
>
>   In regards to pooling size schedules, this is an excellent point! We only compare to a “constant schedule” in Fig. 2, but we could additionally consider a linear schedule (i.e. sf, 2sf/3, sf/3 for instance). We suspect the difference would be marginal, but would definitely include such an ablation in the final paper in the event of acceptance.
>
>   We have not considered any other video transformation strategies, and while this could be interesting to investigate in its own right, we suspect there is more to be gained by utilizing a sufficiently large and diverse set of actual natural videos for training.
>
> - **W5**: Our initial experiments did (of course) include some hyperparameter testing, and in general the results were not particularly sensitive. While time and resource constraints prevent us from conducting a thorough analysis during the rebuttal we would include such an analysis if accepted.
>
> - **Q1**: See our response to W1 above.
>
> - **Q2**: Thank you for this thoughtful question. Layerwise training approaches present the opportunity to reduce the *peak memory demands* of training by operating the the forward and backward pass of each stage sequentially (and freeing the memory associated with the internal activations of each stage after completion) [2]. However the compute requirements have the same complexity as end-to-end training as far as we can tell (and incur some constant complexity overhead associated with computing the intermediate losses). Our procedure (and standard end-to-end learning) are however significantly more compute efficient than adversarial training, which involves a costly nested optimization.
>
> [1] Parthasarathy, Nikhil, Olivier J. Henaff, and Eero P. Simoncelli. "Layerwise complexity-matched learning yields an improved model of cortical area V2." Transactions on Machine Learning Research.
>
> [2] Siddiqui, Shoaib, et al. "Blockwise Self-Supervised Learning at Scale." Transactions on Machine Learning Research.

---

> > ### Author Response · Authors · 2025-11-23
> > **Response to Reviewer QTgA (Part 2)**
> >
> > - **Q3** : See our response to W2 above.
> >
> > - **Q4**: Performance generally saturates for longer videos, though we agree this is a “missing” ablation and would include such an experiment in the final paper in the event of acceptance.
> >
> >   It is worth noting that we believe this saturation is more of a feature of our synthetic video generation procedure, and we would hesitate to make any claims about whether saturation would occur given sufficiently rich dynamics.
> >
> > - We completely agree that this intriguing phenomenon merits further investigation. It is worth clarifying that the V1 frontend, which has fewer frozen stages, does outperform  the V2 frontend on the in-distribution classification task, as would be expected from the number of trainable parameters, but this does not explain the “non-monotonicity” of OOD performance or behavioral alignment with the number of frozen stages. We think it would be helpful to separately consider two "sub-questions":
> >     - Why is the V1 stage inferior to the V2 stage?: As you point out, the classifier here has more tunable parameters and receives inputs containing strictly more information than the V2 stage (due to the information processing inequality), the key question is how the transformation from V1 to V2 exposes or preferentially emphasizes features that facilitate OOD generalization and human like behavior. We think analyzing the geometrical properties of “texture manifolds” (or how they relate to those of “noise manifolds”) would be an excellent starting point!
> >     - Why is the V4 stage inferior to the V2 stage?: It would be necessary to test whether the drop in performance is due to a particular invariance learned at that layer discarding features that are necessary for carrying out human-like categorization, or simply a feature of reducing the number of tunable parameters relative to the V2 stage. We could potentially aim to match the number of non-frozen parameters available for downstream classification on top of each layer by appending additional layers to the classifier that takes the V4 stage as inputs to disambiguate between these two possibilities.
> >
> >   We would be happy to discuss this finding more prominently in the main text, as we certainly feel it is an interesting question for future work that was uncovered somewhat haphazardly in our paper!

---

> ### Comment · Reviewer_QTgA · 2025-11-26
> **Response to Authors' Comments**
>
> Thank you for your comprehensive and thoughtful responses to my review. I appreciate the care you've taken in addressing each point.
>
> **Addressed Concerns:**
>
> I'm particularly pleased to see that you've already implemented the LCL comparison (W3) and included it in the updated manuscript. This was one of my key concerns, and having direct empirical comparisons to other layerwise learning methods significantly strengthens the paper.
>
> Your detailed analysis of the V2 frontend superiority is excellent and provides valuable mechanistic insight. The distinction between the two sub-questions (why V1 < V2 and why V4 < V2) is helpful, and your proposed approaches to disambiguate between invariance effects and parameter count effects are sound. I agree this finding deserves more prominence in the main text.
>
> The preliminary ResNet-50 results (W2) are encouraging, even if they don't fully resolve the architectural generalization question. Your observation that ST-MMCR has more pronounced effects in earlier layers is interesting and suggests the approach may have specific utility for modeling early visual processing.
>
> **Remaining Considerations:**
>
> While I understand the resource constraints during rebuttal, I do encourage you to include the promised ablations (W4, W5) in the final version if accepted. These would aid reproducibility and provide useful guidance for future work building on your approach.
>
> The biological plausibility of the nuclear norm computation (W1) remains a limitation, though I appreciate your acknowledgment and ongoing work on this (Appendix A.6). This caveat should perhaps be stated more prominently in the main text given the paper's biological motivation.
>
> **Updated Assessment:**
> Given your responses, particularly the addition of the LCL comparison and the thoughtful analysis throughout, I am raising my score from 6 to 8. The paper makes solid contributions to biologically-motivated representation learning with strong empirical validation, and the authors have demonstrated good faith in addressing reviewer concerns.
>
> I remain enthusiastic about this work and believe it will be a valuable contribution to the ICLR community.

---

### Official Review · Reviewer_naJ4 · 2025-11-01

**Soundness:** 2
**Presentation:** 3
**Contribution:** 2
**Rating:** 2
**Confidence:** 4

**Summary:**

This paper proposes a novel self-supervised, layerwise learning strategy, termed ST-MMCR, to train deep neural networks as more biologically plausible models of the primate ventral visual stream. The work is motivated by the biological implausibility of global backpropagation and the potential for single, end-to-end objectives to insufficiently constrain internal representations. The authors' method trains the network in stages, applying a local learning objective at each stage based on maximizing multiscale manifold capacity (MMCR). The core novelty is the reapplication of this same loss computation at each stage, arguing this is analogous to the replication of canonical cortical circuits. The model supposedly "matches complexity" by linking this loss to expanding receptive fields (both spatial and temporal) of the network.
The model is evaluated on its ability to predict neural responses in macaque visual areas (V1, V2, V4) and its alignment with human psychophysical data on an out-of-distribution (OOD) object classification task. The results suggest the ST-MMCR model matches or outperforms undefined "architecture-matched baselines" in neural predictivity and shows stronger alignment with human OOD generalization, despite lower standard ImageNet accuracy.

**Strengths:**

- The paper's motivation is strong. It addresses the central challenge of biological plausibility by attempting to replace global backpropagation with a local, layerwise learning signal.

- The idea of a "common computational objective" (MMCR) that is reapplied at each stage, modulated by the expansion of receptive fields is an interesting hypothesis. It aligns conceptually with the biological principle of canonical microcircuits.

**Weaknesses:**

- Problematic and Inaccurate Engagement with Prior Literature: This is a major concern. The paper states that "handcrafted filters that are tuned for orientation and spatial frequency substantially outperform learned models on the V1 dataset." This claim is in direct contradiction with other works in the field. For example, Cadena et al., 2019, PloSCB [1] have definitively shown that DNNs trained on tasks (task-driven models) or end-to-end to predict neural responses (data-driven models) substantially outperform models based on handcrafted Gabor-like filters. Similarly, other literature that cites


- Narrow Framing and Lack of Essential Baselines: The paper almost exclusively cites and builds upon a single prior work [2], while failing to engage with the vast and diverse field of data-driven and task-driven models of the ventral stream. The comparison is limited to "architecture-matched baselines" which are not clearly defined. Crucially, the paper fails to cite or discuss highly relevant work on self-supervised models of visual cortex, such as [3]. That paper also found that task-agnostic, self-supervised objectives can be superior matches to neural data and lead to general-purpose representations. A discussion of how ST-MMCR compares to other self-supervised approaches (e.g., contrastive learning) is a critical and missing piece.


- Superficial Neural Analysis: The analysis is limited to population-level performance metrics. The authors do not leverage their model to investigate why it aligns with the neural data. If the model is interpretable, as implied, the authors should have provided a deeper analysis. For example, can learned properties of model units (e.g., receptive field size, tuning properties) be directly related to the measured properties of the biological neurons they predict? This is a significant missed opportunity to provide mechanistic insight beyond a simple performance number.


- Stimulus Mismatch Between Training and Evaluation: The model is named ST-MMCR, implying "spatio-temporal," and the discussion mentions "temporal pooling duration," suggesting it is pre-trained on videos. However, the model is evaluated on neural data collected from primates viewing static images. The authors must justify why a model trained on temporal dynamics would be expected to produce superior representations for static image processing. This choice complicates any interpretation of the results.


- Surprising Performance of Random Networks: The sparse regression results in Figure 5 are highly counter-intuitive. The fact that a random-weight model achieves respectable performance, particularly in the sparse regime, requires a much deeper explanation. Does this imply that the architecture, pooling, and connectivity itself—rather than the learned weights—are responsible for a large portion of the alignment?

- Biological Plausibility of the Loss Function Itself: This is a central contradiction. The authors admit in the discussion that "the final MMCR loss, which relies on nuclear norm computation, does not currently have an obvious counterpart in the world of biological modeling." However, this undercuts the paper's primary claim of biological plausibility. If the learning signal itself is biologically implausible, then replacing global backpropagation with a local but equally implausible computation is not a step forward.


- Vagueness of "Manifold Capacity": The paper's central mechanism, "multiscale manifold capacity," is never clearly defined. It is unclear what these "manifolds" are. Are they manifolds of object classes? Image augmentations? Stimulus identity across time? Without a concrete definition, "manifold capacity" remains an abstract mathematical construct that is difficult to connect to a clear computational goal of the visual system.

[1] https://journals.plos.org/ploscompbiol/article?id=10.1371/journal.pcbi.1006897


[2] https://arxiv.org/abs/2312.11436


[3] https://journals.plos.org/ploscompbiol/article?id=10.1371/journal.pcbi.1011506

**Questions:**

1. Could the authors please address the claim that "handcrafted filters... outperform learned models"? This appears to directly contradict findings from Cadena et al. (2019) and others. Can you clarify which V1 dataset and handcrafted models are being referenced that produce this result?


2. Why was the analysis limited to population-level predictivity? Can the authors provide any analysis linking learned model parameters or unit properties (e.g., RF size, feature tuning) to the corresponding properties of the biological neurons they predict?


3. What is the justification for using video (spatio-temporal) data to pre-train a model that is then evaluated on neural responses to static images? Have the authors tested whether training on a large dataset of static images with the same MMCR loss yields comparable or better results?


4. How do the authors interpret the surprisingly high performance of the random-weight network in the sparse regression analysis? What does this imply about the relative contributions of architecture versus learned weights in explaining neural responses?


5. Given that the nuclear norm computation is admittedly not biologically plausible, what is the authors' view on the model's contribution? Are there known (or hypothetical) local, Hebbian-like or dynamic circuit computations that could approximate the maximization of nuclear norm, thereby bridging this plausibility gap?


6. Could the authors please provide a concrete definition of the "neural manifolds" that the MMCR objective is separating? What specific stimulus properties, transformations, or classes define these manifolds at each stage of the hierarchy?

---

> ### Author Response · Authors · 2025-11-23
> **Response to Reviewer naJ4 (Part 1)**
>
> Thank you for your thorough review of our submission, we hope that our responses show there is significant opportunity to improve the clarity of the contribution and presentation of our work. Below we respond point by point to each of the listed weaknesses and questions:
>
> - **W1**: By “on the V1 dataset” we meant to restrict our claims to the scope where we feel confident they are true (the public BrainScore dataset).  Figure 8 of Cadena et al. 2019 shows that (for their own neural dataset) that a Gabor filter bank explains about 10% less variance than a pretrained VGG or a CNN directly optimized to predict responses. But for the Freeman&Ziemba (BrainScore) dataset, the situation is the opposite: handcrafted steerable pyramid filters have roughly 10% higher predictivity than trained AlexNets  (see also our detailed response to reviewer fHVb for a table containing the top-5 models on this V1 dataset, which shows the v1-pyr-nodown model which as a top performing model). Nevertheless, we do agree that a broader discussion of these differences is warranted and we will update the text of the paper accordingly.
>
> - **W2**:
>     - By “architecture-matched”, we simply mean that we consider models with identical architecture, so as to focus on the effects of objective and training procedure.
>     - The prior publication [2] showed comparison to other self-supervised models, finding that they had performance similar to standard supervised networks.  We have now compared the performance of Barlow twins [4] as well as directly with [2], which are shown in Figs. 3,4&6 (we are currently working on adding the same comparisons to Fig. 5 ).  Performance of our model is better than end-to-end self supervised training and comparable to [2], the prior SOTA.
>     - We felt that the Zhuang 2021 paper (line 37) was more appropriate as it considers responses in Macaque rather than mouse. However we would be happy to include a broader discussion of both papers in the revised text.
>
> - **W3**: We certainly agree that comparing the tuning properties of real and model neurons can provide a more interpretable signal as to the degree of model alignment. However, although the attributes that govern tuning  in V1 (orientation, spatial frequency, phase, RF size and location) are widely accepted, they are less informative for later stage neurons (V2, or V4), and surely do not capture the more interesting (nonlinear) response properties that differentiate these later stages from V1. The texture and spectral noise responses properties measured as a part of the “Marques2020_V1” benchmark in the BrainScore framework provide a first step, as they were used to differentiate V1 from V2 responses. We are actively working on an implementation of this comparison (and hope to have results to share before the end of the discussion period).
>
> - **W4**: We agree that our choice to learn a representation of static images from the structure of dynamic visual inputs is non-standard and required more motivation than it was given in the text. The high-level motivation comes from the “two-stream” or “what/where” hypothesis which posits that the ventral stream aims to provide a representation that is  relatively invariant to temporal changes while the dorsal stream learns to represent dynamical aspects of the world (i.e. motion) in a content agnostic way. This motivation is shared with some classic works in computational neuroscience, i.e. [5, 6]. Consistent with this framing, the majority of recordings in the ventral stream present static stimuli, which are therefore appropriate for the comparisons we make.
>
>   As we train our model to be invariant to temporal dynamics (at different scales across different stages), we would expect it to be unable to account for aspects of neural responses that encode for such information, and which would be present  in datasets where video stimuli are presented (and would be much more prevalent in the dorsal stream).
>
>
> [4] Zbontar, Jure, et al. "Barlow twins: Self-supervised learning via redundancy reduction." International conference on machine learning. PMLR, 2021.
>
> [5] Földiák, Peter. "Learning invariance from transformation sequences." Neural computation 3.2 (1991): 194-200.
>
> [6] Wiskott, Laurenz, and Terrence J. Sejnowski. "Slow feature analysis: Unsupervised learning of invariances." Neural computation 14.4 (2002): 715-770.

---

> > ### Author Response · Authors · 2025-11-23
> > **Response to Reviewer naJ4 (Part 2)**
> >
> > - **W5**: We agree that the performance of random networks on neural predictivity is somewhat counterintuitive. We interpret it as an indication that the model architecture imposes constraints that are well-matched to the neurons.  Specifically, the receptive field sizes (that arise from the size and cascading of filters), and the complexity of nonlinearities (which are governed by the number of cascaded nonlinear rectification operations) are approximately matched to those of the cells.  This is true even without training (i.e., with random weights), but in that case, one needs to interpolate over many artificial neurons to achieve a good match to each real neuron. The comparison allows us to isolate these effects from the effects of training (objective and dataset).  This interpretation is supported by many publications, for example:
> >     - "Convolutional architectures are cortex-aligned de novo" [7]: This paper  investigates the impact of  embedding dimensionality and other architectural elements on the predictivity of untrained networks. See Fig. 2, which shows that the performance of randomly initialized convolutional  networks (1) explain a non-trivial amount of variance in neural recordings across measurement modalities and brain areas, even when using a small number of dimensions and (2) with sufficiently large dimensionality said  networks approach the predictivity of their trained counterparts.
> >     - "Neural responses in early, but not late, visual cortex are well predicted by random-weight CNNs with sufficient model complexity" [8]: This paper explores similar questions to the one described above. See Fig. 1 which compares random and trained networks on a different macaque V1 dataset (Cadena et al. 2019’s) as well as on human fMRI measurements and similarly observes that random networks perform very similarly to their trained counterparts in explaining the responses, especially in early cortical areas.
> >     - "Unsupervised neural network models of the ventral visual stream" [9]: See Fig. 2, in V1 and V4 the untrained baseline model achieves between 80-90% of the performance of a model trained with standard object recognition supervision.
> >
> > - **W6**: We respectfully disagree.  Achieving equal performance with a local layerwise learning rule is a significant and sensible first step toward biological plausibility. Our paper does this.  That said, we certainly agree that there is more work to be done, and we have begun work on a plausible circuit-based local optimization algorithm for our model. We shared a sketch of this argument in a new appendix section of the manuscript: A.6 A Circuit for Maximizing the nuclear norm.
> >
> >   In brief, we leverage techniques  for deriving dynamic neural circuits that optimize objective functions found in the “similarity matching” literature [10], and the fact that the nuclear norm of a matrix can be written as the solution of an optimization problem. We note that this appendix section is very much a “work in progress,” and is far from constituting a complete solution to the problem. We hope that our inclusion of this early-stage work helps to convey that nuclear norm optimization is not inherently implausible, and we would be happy to integrate description of these ideas should the paper be accepted.
> >
> > [7] Kazemian, Atlas, Eric Elmoznino, and Michael F. Bonner. "Convolutional architectures are cortex-aligned de novo." Nature Machine Intelligence (2025): 1-11.
> >
> > [8] Farahat, Amr, and Martin Vinck. "Neural responses in early, but not late, visual cortex are well predicted by random-weight CNNs with sufficient model complexity." bioRxiv (2025): 2025-02.
> >
> > [9] Zhuang, Chengxu, et al. "Unsupervised neural network models of the ventral visual stream." Proceedings of the National Academy of Sciences 118.3 (2021): e2014196118.
> >
> > [10] Pehlevan, Cengiz, Anirvan M. Sengupta, and Dmitri B. Chklovskii. "Why do similarity matching objectives lead to Hebbian/anti-Hebbian networks?." Neural computation 30.1 (2017): 84-124.

---

> > > ### Author Response · Authors · 2025-11-23
> > > **Response to Reviewer naJ4 (Part 3)**
> > >
> > > - **W7**: Thank you for pointing out this opportunity to improve the clarity of our work. In the setting of this paper, a “manifold” is a spatiotemporally local group of stimuli (or most precisely, the neural representation of such a group of stimuli, as detailed in Fig. 1). The size of the neighborhood in space and time membership to a single manifold grows between successive stages of the representation, with the earliest stages considering only short and relatively low field of view inputs as individual manifolds, and with the final stage considering stimuli in a larger neighborhood in space and time to belong to the same manifold. I.e. manifolds are points sharing the same “stimulus identity” across space and time, but the extent of space and time that define a manifold grow along the hierarchy. This is distinct from the setting of [11] where a single manifold corresponded to many possible augmentations of a single image, and markedly different from the case where a single manifold consists of stimuli with the same class label (in which case maximizing the capacity would be very similar to supervised training). With manifolds defined as such, maximizing the manifold capacity is a particular form of coding efficiency.
> > >
> > >   We would be happy to modify the introduction of the text in order to help convey this more effectively. If this issue is still unclear please let us know. Besides the issue of defining a manifold, the definition of “manifold capacity,” is also only briefly mentioned and, in retrospect, this is likely insufficient. Would it perhaps be helpful if we added appendices describing the mathematical definition of manifold capacity in general?
> > >
> > > - **Q1**: We are referring to the Freeman and Ziemba V1 dataset and the steerable pyramid features used in our paper, which outperforms the learned models with an AlexNet architecture we consider. Furthermore on the brainscore leaderboard for this dataset this handcrafted model is ranked 4th overall out of several hundred submissions, only outperformed by one model with handcrafted inductive biases and two  networks using adversarial training. Please see also our response to W1 above.
> > >
> > > - **Q2**: Please see our response to W3 above.
> > >
> > > - **Q3**: We hope our response to W4 speaks to this question. In terms of training on only static images using a similar framework,  this is very similar to what is accomplished by the LCL model of Parthasarathy et al. 2023. We have added direct comparisons with this approach in the Figures of the revised manuscript. In short, this does provide very similar results in terms of neural predictivity! One of the core contributions of this work is to show that, given the appropriate modifications, one can achieve strong predictivity while using a more biologically plausible learning algorithm. See also our response to W1 of Reviewer  fHVb, where we detail the changes between LCL and ST-MMCR explicitly.
> > >
> > > - **Q4**: Please see our response to W5 above.
> > >
> > > - **Q5**: Please see our response to W6 above.
> > >
> > > - **Q6**: The concrete definition of a neural manifold is: the set of neural  responses at layer $l$ generated by a stimulus of spatial extent $s_l$ and temporal duration $t_l$ (i.e. it is a “point cloud” manifold). Notably $s_l$ and $t_l$ are different at different layers $l$ (they grow larger in size at each successive stage).
> > >
> > >
> > >
> > > [11] Yerxa, Thomas, et al. "Learning efficient coding of natural images with maximum manifold capacity representations." Advances in Neural Information Processing Systems 36 (2023): 24103-24128.

---

### Official Review · Reviewer_EJNc · 2025-11-01

**Soundness:** 3
**Presentation:** 3
**Contribution:** 2
**Rating:** 6
**Confidence:** 4

**Summary:**

Present an unsupervised method with layer-by-layer training (rather than end-to-end training) by optimizing a previously suggested measure of manifold capacity. A smart choice of the scaling in each layer, which corresponds to the scale of the receptive fields, allows for representing the multiscale nature of visual stimuli. A new method for training on ImageNet-derived video-like stimuli is used. The resulting method is competitive with other methods in terms of predicting responses along the mammalian visual system, and also in terms of the classification accuracy achieved, despite being unsupervised.

**Strengths:**

* An unsupervised method for layer-by-layer training.
 * Achieving interesting results in both accuracy and prediction of neural responses.
 * An interesting method for training a static model using "synthetic videos", derived by smoothly sampling still images from the ImageNet dataset, where training is done over the "temporal responses" of the model.

**Weaknesses:**

* Insufficient baselines: the chosen baselines for Figures 3-6 are somewhat arbitrary. The authors could have compared with "a simple unsupervised method", as well as "SOTA unsupervised method", thus placing the suggested method in between in terms of the range of unsupervised methods. The authors could have compared with "a supervised version of the same architecture", or "the same architecture without training on video-like stimuli", thus making the conclusions much clearer (see next point).
 * It is unclear how much of the results should be attributed to (i) the loss function used; (ii) the training method used; (iii) the stimuli used for training; or (iv) the smart choice of the scaling. For example, it might be that the good predictability of neural data is achieved only through the smart scaling from layer to layer.

**Questions:**

* Can you suggest what part of the results presented should be attributed to (i) the loss function used; (ii) the training method used; (iii) the stimuli used for training; or (iv) the smart choice of the scaling?

---

> ### Author Response · Authors · 2025-11-23
> **Response to Reviewer EJNc**
>
> Thank you for your positive review of our paper. Below we respond point by point to each weakness and question:
>
> - **W1**: We agree that careful comparisons to appropriate baselines are necessary to clearly demonstrate the value of our approach. As such we have added a comparison to what is, to our knowledge, the SOTA unsupervised method (LCL - Parthasarathy et al. 2024), as well as a comparison to a standard unsupervised method (Barlow Twins). These comparisons appear in Figs. 3, 4, and 6 of the updated manuscript, and will be added to Fig. 5 in the final version of this paper. We also directly compare to a supervised version of the same architecture (this is the “Supervised” model across Figs. 3,4,&5) and we will make this clearer in the revised text.
>
> - **W2**: We agree, but rather than separately vary the effects of all four of these ingredients, we aimed to freeze the architecture and dataset, so as to focus exclusively on modifications to the objective and training procedure.  We do demonstrate that a large amount of neural predictivity is simply a result of the inductive bias of the chosen architecture, as demonstrated (figs. 3, 5) by the performance of our random baseline model (this observation is also supported in [1, 2]). Moreover, a large amount of neural predictivity gained by optimization can be attributed to the scale and diversity of the training set [3]).
>
>   We suspect that much of the neural predictivity gains induced by our method (relative to the baselines) arise from the appropriate selection of task difficulty (i.e. the total scale of variation over the synthetic video clips, or the “augmentation strength” in a more standard SSL parlance). We did attempt to disambiguate these effects. For example, in Fig. 1 we compare to a naive global scaling strategy, and in Fig. 8 we ablate the layerwise aspects of our training strategy, and would endeavor to explicitly test this hypothesis via an ablation should the paper be accepted (time and resource constraints prevent this experiment from being conducted during the rebuttal period).
>
> - **Q1**: See our response to W2 above.
>
> [1] Kazemian, Atlas, Eric Elmoznino, and Michael F. Bonner. "Convolutional architectures are cortex-aligned de novo." Nature Machine Intelligence (2025): 1-11.
>
> [2] Farahat, Amr, and Martin Vinck. "Neural responses in early, but not late, visual cortex are well predicted by random-weight CNNs with sufficient model complexity." bioRxiv (2025): 2025-02.
>
> [3] Conwell, Colin, et al. "What can 1.8 billion regressions tell us about the pressures shaping high-level visual representation in brains and machines?." BioRxiv (2022): 2022-03.

---

> > ### Comment · Reviewer_EJNc · 2025-11-25
> > **Response to authors**
> >
> > I appreciate your response, especially the addition of several important baselines, which help me better appreciate your contribution. The main limitation of the manuscript is that it achieves improvement over these baselines by combining several ideas: different training procedure [data augmentation method], different architecture [pooling structure], and different loss [MMCR]. This criticism is shared by an additional reviewer (fHVb, who was less generous in terms of rating than me), and I don't think your improvements shed light on how important those different aspects are to the reported result. The fact that using MMCR loss alone is not enough to achieve improvement is suggestive that you have not yet captured the necessary ingredients for layerwise training.

---

### Official Review · Reviewer_fHVb · 2025-11-02

**Soundness:** 2
**Presentation:** 3
**Contribution:** 1
**Rating:** 2
**Confidence:** 3

**Summary:**

This submission proposes a local learning appraoch in deep networks based on manifold capacity. The method relies on the change in the receptive field size of the network across layers and applies MMCR loss at different layers independently. The proposed method relies on methods from previous papers, most importantly Yerxa et al. 2023 and Parthasarathy et al. 2024. The empirical results show that the proposed method does better than other baselines in predicting neural activity in areas V1,V2, and V4 of monkeys.

**Strengths:**

1. the proposed method is more biologically plausible.
2. the figures were clear and easy to understand
3. the paper was well written

**Weaknesses:**

Overall there were several issues that affected my impression of its Soundness and Contribution.

1. the specific contribution of this submission and what factors distinguished it from some of the related work was unclear. I’m not sure how the contribution of this paper is different from that in Parthasarathy et al. 2024. My understanding is that this paper has the same approach as that paper but the implementation is slightly different to improve biological plausibility. Is this true? Please explain in more detail what the differences are and why they warrant considering the present work as novel and impactful.

2. Some of the choices made in the model were not well justified/explained.

- what is the reason for restricting the network to only 3 stages? The proposed approach seems to be general, why not trying it in a deep architecture? (line 64)

- also see my questions

3. Additional experiments needed.

- the synthetic dataset is somewhat unconventional (line 154). A version of the model trained on natural videos would be very informative both in terms of the importance of the naturality of the videos but also scalability of the method.

- given the close relationship between the current work and Parthasarathy et al. 2024, I was surprised to see that it was not considered as a baseline for the main results.

4. Unclear/unjustified statements

- line 55. Typically there are more artificial neurons in the first stages of processing in CNNs due to larger spatial dimensions. It’s unclear in what sort of neural network model this statement is true for.

- line 170. please provide supporting evidence for this statement. Several work from James DiCarlo including BrainScore has shown that deeper models such as ResNet50 are better than shallower ones like Alexnet at predicting brain activity across regions.

- alexnet has 9 layers, but the model is said to use Alexnet architecture but only three stages. Please explain more clearly what’s the relation to Alexnet exactly. Is it adopting the first few layers of that architecture?

- it was unclear what the experiments in section 3.1 revealed. The opening of that section speaks about previous literature suggesting that models with similar architectures and training datasets share representations. Looks like the results in fig 4 support that view. Is that so? I didn’t see that discussed

5. Statistical significance.

- values reported in figure 2 and 8 are extremely close to each other. Have you repeated the analyses multiple times? Are the differences statistically significant? Similar question for the main results in fig 3 and 5

6. Others

- Line 67, acronym used before being defined

**Questions:**

- what kind of pooling operation is used? line 225

- I believe BrainScore also includes several neural datasets from V1 and V2. Have you tested using that data as well? Is the data considered here better or larger than that?

- Please explain more clearly whether the global average pooling strictly considered to calculate the loss or it affects the output generated by each layer?  line 267

- given the results showing improved OOD generalization, does the model perform better against adversarial attacks as well?

---

> ### Author Response · Authors · 2025-11-23
> **Response to Reviewer fHVb (Part 1)**
>
> Thank you for your thorough review of our work. Please find our point by point responses to the listed weaknesses and questions below:
> - **W1**: Our approach was directly inspired by  Parthasarathy et al. 2024, and indeed our central methodological contribution is demonstrating that it is possible to modify the structure of inputs such that all stages are trained on a single coherent stimulus (**Modification 1: synthetic videos vs. stage specific static augmentations**), by making appropriate changes to the architecture (**Modification 2: Spatially and Temporally Local Pooling vs. Global Spatial Pooling**) and training objective (**Modification 3: Use of MMCR objective which has constant complexity in the number of positive samples vs. Barlow Twins which has quadratic complexity**). Each of these modifications are synergistic, and together enhance the biological plausibility of our approach relative to Parthasarathy et al. 2024.. In addition, the Parthasarathy paper only considered a 2-stage network corresponding to visual areas V1/V2. We’ve used a 3-stage architecture, which yields representations that are predictive of a deeper brain layer (V4).  We feel that these contributions are each novel and warrant inclusion in the literature. In the case the paper is accepted we would augment the text of section 1.1 to more explicitly express this sentiment.
>
> - **W2**: Our network design choice is explicitly geared towards developing minimal models of neural responses in visual cortex. While it is certainly interesting to consider deeper and more expressive architectures (for example, to mimic responses in area IT), this approach is already very well represented in the literature. As such we take a different perspective: what is the simplest architecture and training scheme that is biologically plausible (local computation and learning) that can produce predictive representations in mid-ventral areas V2 and V4?
>
>     Determining the extent to which our approach can be directly adapted to the setting of deeper networks is an excellent topic for future research. Doing so would require a careful consideration of how expressivity scales with depth in more complicated architectures, as a key to our approach is to match the task difficulty with how flexible the representation at a given stage is. Furthermore, this would likely require either enriching the synthetic video generation process, or making the move to true natural videos for training. This is because the image resolution and size of spatial transformations we apply are already quite large (see Table 1 in Appendix A.2).
>    It is perhaps worth mentioning that in the early stages of the development of this project we applied the ST-MMCR loss as a regularizer alongside a standard end-to-end supervised learning signal in a deep network (ResNet-50) trained on object recognition (ImageNet-1k). In this setting we observed that the additional loss had a large effect on the representation at early stages, and provided little additional constraint on the representations at latter stages of the network. These results motivated  us to investigate the proposed loss as a stand alone objective in a relatively simple architecture. We would be happy to include some of these experiments and implementation details in an appendix section, but opted not to as the setting considered differed significantly from the core set of results.
>
> - **W3A**: We agree that applying our approach to actual natural videos is an important next step. We’ve used ImageNet+synthetic videos because it has previously been demonstrated that the scale and diversity of a dataset are critical to learning strong image based representations from video, and  inducing neural predictivity via optimization (see [1, 2]). As such we hoped to facilitate fairer comparisons between our approach and existing work by fixing the data diet to a standard and common choice.
>
> - **W3B**:  The weights obtained by Parthasarathy et al. are not publicly available, which is why we did not include comparisons in the initial submission. We have since developed our own implementation to reproduce the core results of that paper and include comparisons in the updated manuscript's Figs. 3, 4, & 6 (this is the "LCL" model in these figures; we are working on adding it to Fig. 5 as well).
>
> [1] Conwell, Colin, et al. "What can 1.8 billion regressions tell us about the pressures shaping high-level visual representation in brains and machines?." BioRxiv (2022): 2022-03.
>
> [2] Parthasarathy, Nikhil, et al. "Self-supervised video pretraining yields robust and more human-aligned visual representations." Advances in Neural Information Processing Systems 36 (2023): 65743-65765.

---

> > ### Author Response · Authors · 2025-11-23
> > **Response to Reviewer fHVb (Part 2)**
> >
> > - **W4A**: Thank you for pointing out our poor communication on this point. We were simply referring to the fact that early stages of convolutional networks tend to have fewer channels or neurons with distinct spatial tuning profiles. You are certainly correct that the total dimensionality of early layer feature maps tends to be larger as a result of the downsampling stages in typical implementations.
> >
> > - **W4B**: Our interpretation of the public BrainScore leaderboard (as well as prior results in the literature [3]) is that deeper networks enjoy a pronounced advantage over shallower counterparts in their ability to predict responses in deeper cortical areas, especially in area IT. See below the BrainScore leaderboards Top 5 for Area V1 and V2, considering either only the datasets we also employed or additional fMRI predictivity benchmarks as well.
> >
> >   Note that for the electrophysiology datasets we considered for these two areas  (which are the only such datasets in BrainScore), the following statements are true:
> >     - Two of the top 5 models for area V1 use handcrafted features/inductive biases, and the pyramid baseline we use in our paper (v1-pyr-nodown) ranks 4th overall, with a score that is ~95% of the best overall model (which also uses handcrafted inductive biases in training).
> >     - In area V2 the robust alexnet architecture is ranked 3rd overall with a score that is ~99% of the top ranked model
> >
> > Furthermore, when considering all neural datasets (including the additional fMRI dataset), the top 5 performers for both areas use Alexnet architectures. Finally it is worth noting that while the shallower Alexnet architecture (when paired with adversarial training) appears near the top of the BrainScore leaderboard for these V1 and V2 datasets, these models tend to perform poorly at predicting measurements in IT (often ranking outside the top 300 on benchmarks in this area).
> >
> > | Model Rank   | Freeman & Ziemba V1 | V1 | Freeman & Ziemba V2 | V2 |
> > | --------------- | ------- | --- | ------- | --- |
> > | 1.     | resnet50_primary_visual_cortex: 0.43 | **alexnet_training_seed_01**: .61 | resnet50_robust_l2_eps3: .40 | AlexNet_SIN: .50
> > | 2.     | [fulltest_microblock_robust] : 0.43      | **alexnet_training_seed_10**: .60 | FrankRobWobv0: .40.              | **alexnet_training_seed_01**: .43
> > | 3.     | [tf_efficientnet_robust]: 0.41                | **alexnet_training_seed_07**: .60 | **alexnet_robust**: .40            | **alexnet_training_seed_10**: .41
> > | 4.     | **v1-pyr-nodown**: 0.41                      | **alexnet_training_seed_09**: .59 | resnet50_robust_l2_eps1: .39 | **alexnet_training_seed_07**: .40
> > | 5.     | [resnet50_cutmix_robust]: 0.39           | **alexnet_training_seed_02**: .57 | voneresnet-50-robust: .39      | **alexnet_training_seed_03**: .60
> >
> > - **W4C**: Yes, our model adopts the first several layers of the AlexNet architecture. We agree that this needs to be more clearly elaborated in the main text ( section 2.2), with more explicit reference to Appendix A.3 where the architecture is defined in detail.
> >
> > - **W4D**: The key takeaways from the experiments of Section 3.1 are:
> >     - Certain types of images induce responses that are better predicted by deep networks, and the properties of these images vary across brain areas. In V1, spectral noise responses are marginally more predictable than those for their texture counterparts, and the reverse is true in area V2. In V4 more  texturized images evoke less aligned responses than natural images.
> >     - The predictivity of random networks converges much more slowly than that of trained networks (i.e., in the V1 and V2 panels, the random network curve gradually approaches the trained network as the mapping becomes more and more flexible).
> >   These results are generally in line with your observation: both architecture and training dataset are important for shaping a particular representation. We feel these results strengthen the evidence for this observation in a way that is novel to the literature, by breaking down the score across different “partitions” of the dataset.
> >
> > [3] Parthasarathy, Nikhil, Olivier J. Henaff, and Eero P. Simoncelli. "Layerwise complexity-matched learning yields an improved model of cortical area V2." Transactions on Machine Learning Research.

---

> > > ### Author Response · Authors · 2025-11-23
> > > **Response to Reviewer fHVb (Part 3)**
> > >
> > > - **W5**: We have repeated our default training procedure starting from 10 different random initializations and have found our results to be robust. We have updated Fig. 2 to include error bars indicating variability over random seeds (note that this is slightly different than the error bars of Figs. 3 and 4 which indicate variability over train-test splits in neural evaluation.
> > >   A correct (if somewhat cautious)  interpretation of Fig. 8 is that our simplified credit assignment procedure does not have an impact on neural predictivity. We believe that this null  result is still relevant to/bolsters our contributions, as we explicitly demonstrate that we did not leave any predictivity “on the table” by taking a more biologically plausible approach. We will modify lines 354-355 in the manuscript to make this stance more explicit in the main text.
> > >
> > > - **Q1**: The operator LSP(s) performs a spatial average over a “central” square (of size s) in a 2-D feature map (channel).  This is applied independently across channels.
> > >
> > > - **Q2**: The V1 and V2 datasets we consider are the (only) macaque electrophysiology datasets on BrainScore for these areas. There is also one human fMRI dataset  for these areas (that was recently added to the BrainScore platform) and one additional V1 benchmark in BrainScore that contains derived receptive field properties of (e.g., distribution of preferred spatial frequencies), but since it is not a neural response regression dataset, we did not include it.
> > >
> > >   These datasets have (in part due to the BrainScore leaderboard) become defacto benchmarks in the field.  We’ve implemented our own evaluations/regressions on these, allowing us to perform more elaborate comparisons (e.g., the sparse regression experiments) rather than restricting to a single score per model per dataset.
> > >
> > > - **Q3**: The pooling mechanisms discussed in this portion of the paper are strictly used to calculate the loss, and do not affect the inputs to subsequent downstream layers. We will improve the clarity of the text that describes this.
> > >
> > > - **Q4**: This is an interesting question. However, due to time and resource constraints we will not be able to investigate the adversarial robustness of our model during the rebuttal period (which is a computationally intensive evaluation given  that our models operate on 224x224 inputs). We are not aware of any definitive statements in the literature that OOD generalization and adversarial robustness are strongly correlated.

---

> > > > ### Comment · Reviewer_fHVb · 2025-11-26
> > > >
> > > > Thank you for responding to my comments. You have addressed weaknesses 3-5 to the mots part. However, after reading your response to W1 and W2, I still feel that this work is too close to prior recent work and the experiments should be extended to regular deep models to better assess its functional merit and beyond biological plausibility.
> > > >
> > > > I also do agree with reviewers EJNc and naJ4 that a more comprehensive set of baselines would be needed to give the readers a better view of where this contribution stands. Also with reviewers naJ4 and QTgA on the point that the framing about biological plausibility is evidently at odds with the reality.

---

### Meta-Review · Area_Chair_AiVu · 2025-12-15

**Summary:**

This work proposes a local layer-wise self-supervised learning (SSL) method to maximally separate image manifolds wrt spatial-temporal variations, producing representations predictive to visual cortex recordings. The reviewers recognize the novel application of MMCR as a stage-wise SSL loss with interesting connections to neural science. The major concerns are the the novelty and positioning among existing literature (Parthasarathy et al, etc.) and lack of essential baselines. The authors' rebuttal does not fully address the concerns, which require extensive ablation studies and broader comparisons (and therefore a major revision). I recommend rejection.

**Reviewer Concerns:**

The reviewers fHVb, EJNc, naJ4 have major concerns on the overall novelty and positioning. The reviewers QTgA, EJNc, fHVb, naJ4 are concerned about missing baselines, and related ablation/sensitivity studies. These are not fully addressed in the rebuttal. Reviewers fHVb and EJNc confirmed before the shutdown that their concerns are still outstanding.

**Reviewer Scores:**

The reviewers naJ4 and fHVb would not increase their scores based on the conversation and rebuttal.

The most positive reviewer QTgA could increase the score. However this increase alone is not sufficient to flip the score to the acceptance side.

The EJNc would not increase their score and may remain borderline.

---

### Decision · Program_Chairs · 2026-01-26

Reject